# A natural mutation in the promoter of *Ms-cd1* causes dominant male sterility in *Brassica oleracea*

Fengqing Han[1,6], Kaiwen Yuan[1,6], Wenru Sun[1,2,6], Xiaoli Zhang[3], Xing Liu[1], Xinyu Zhao[1], Limei Yang[1], Yong Wang[1], Jialei Ji[1], Yumei Liu[1], Zhansheng Li[1], Jinzhe Zhang [1], Chunzhi Zhang [4], Sanwen Huang [4,5], Yangyong Zhang [1] ✉, Zhiyuan Fang [1] ✉ & Honghao Lv [1] ✉

Male sterility has been used for crop hybrid breeding for a long time. It has contributed greatly to crop yield increase. However, the genetic basis of male sterility has not been fully elucidated. Here, we report map-based cloning of the cabbage (*Brassica oleracea*) dominant male-sterile gene *Ms-cd1* and reveal that it encodes a PHD-finger motif transcription factor. A natural allele *Ms-cd1_{PΔ−597,}* resulting from a 1-bp deletion in the promoter, confers dominant genic male sterility (DGMS), whereas loss-of-function *ms-cd1* mutant shows recessive male sterility. We also show that the ethylene response factor BoERF1L represses the expression of *Ms-cd1* by directly binding to its promoter; however, the 1-bp deletion in *Ms-cd1_{PΔ−597}* affects the binding. Furthermore, ectopic expression of *Ms-cd1_{PΔ−597}* confers DGMS in both dicotyledonous and monocotyledonous plant species. We thus propose that the DGMS system could be useful for breeding hybrids of multiple crop species.

Crop improvement via heterosis has long been applied to increase crop yield, adaptability, and resistance to biotic and abiotic stresses[1,2]. During the past few decades, heterosis utilization has revolutionized plant breeding, resulting in 3.5–200% yield increases in different crop species[3].

Male sterility is the most effective tool for heterosis utilization[3,4]. Genetically, there are two types of male sterility: cytoplasmic male sterility (CMS) and genic male sterility (GMS)[5]. CMS and thermo-sensitive/photoperiod-sensitive genic male sterility (TGMS/PGMS) have been widely applied in hybrid breeding[3]. However, these approaches suffer from several intrinsic problems. For instance, the CMS lines exhibit low hybrid seed quality and yields, low genetic diversity, and a risk of increased disease susceptibility. The TGMS/PGMS system, on the other hand, is susceptible to fluctuating climate conditions and can generate low-purity hybrid seeds[3,6]. Dominant genic male sterility (DGMS) is considered as a better approach for efficient utilization of heterosis[4], considering it increases nitrogen use efficiency by allopollination (a 1:1 mixture of male-fertile and male-sterile hybrids) and is feasible for pyramiding multiple genes[4,7,8]. Therefore, identification of genes controlling DGMS phenotype and developing practical male sterility systems are important for hybrid seed production.

[1]State Key Laboratory of Vegetable Biobreeding, Institute of Vegetables and Flowers, Chinese Academy of Agricultural Sciences, Beijing 100081, China. [2]Key Laboratory for Vegetable Biology of Hunan Province, Engineering Research Center for Horticultural Crop Germplasm Creation and New Variety Breeding, Ministry of Education, Hunan Agricultural University, Changsha 410128, China. [3]State Key Laboratory of Vegetable Biobreeding, Tianjin Academy of Agricultural Sciences, 300192 Tianjin, China. [4]Shenzhen Branch, Guangdong Laboratory of Lingnan Modern Agriculture, Genome Analysis Laboratory of the Ministry of Agriculture and Rural Affairs, Agricultural Genomics Institute at Shenzhen, Chinese Academy of Agricultural Sciences, Shenzhen, Guangdong 518120, China. [5]Chinese Academy of Tropical Agricultural Sciences, Haikou, Hainan 571101, China. [6]These authors contributed equally: Fengqing Han, Kaiwen Yuan, Wenru Sun. ✉e-mail: zhangyangyong@caas.cn; fangzhiyuan@caas.cn; lvhonghao@caas.cn

*Brassica oleracea* has been cultivated for more than 3000 years. This species includes several economically important vegetable crops (known as cole vegetables), such as cabbage, cauliflower, broccoli, Brussels sprouts, kohlrabi and kale[9]. Cole vegetables have played an important role in the world food supply for centuries[10]. They were cultivated in an area of approximately 3.8 million hectares in 2021, with a total yield of 97.6 million tons, accounting for 8.32% of global vegetable production (Food and Agriculture Organization; https://www.fao.org). In addition, *B. oleracea* is a precursor of *B. napus* and *B. carinata*, and thus represents an important germplasm for the improvement of oilseed crops[11].

Ogura CMS-based system has been widely applied in hybrid seed production in *B. oleracea*[12]. However, similar to most CMS systems, Ogura CMS lines exhibit floral bud abnormalities, reduced flower size, low hybrid seed quality and yield, low genetic diversity, and changes in important quality traits[13,14]. The dominant male-sterile mutant of *B. oleracea* 79-399-3 (whose causal locus was named *Ms-cd1*), was identified from a spring cabbage population[15]. It has normal-sized flowers and anthers but cannot produce viable pollen[15]. Using 79-399-3 as donor, DGMS lines of cabbage, broccoli, kohlrabi, and Chinese kale have been subsequently generated for their respective hybrid breeding[16]. The male sterility of these DGMS lines is quite stable in diverse genetic backgrounds; however, in very rare cases, trace amounts of pollen grains can be produced from a few anthers of *Ms-cd1* heterozygous plants (HE-DGMS), enabling the production of homozygous male-sterile lines (HO-DGMS) from selfing progenies[15]. Based on this DGMS, commercial seed production systems have been developed for multiple cole vegetables[15]. Previous studies have reported molecular markers associated with *Ms-cd1* and this gene was found to be located on C09[16,17]. However, the gene responsible for this DGMS phenotype remains unknown.

In this study, we report map-based cloning of the cabbage dominant male-sterile gene *Ms-cd1*, explore the underlying molecular mechanism leading to male sterility, and propose the utilization of the system for hybrid breeding in multiple crop species. We envision that the cloned *Ms-cd1* gene and its associated DGMS system can help to improve the breeding of hybrid cultivars.

## Results

### Dominant male-sterile phenotype of *Ms-cd1* plants

Plants of the *Ms-cd1* dominant genic male-sterile DGMS01-20 line have normal-sized anthers but fail to produce viable pollen (Fig. 1a, b). Similar to *Arabidopsis*[18], anther development of cabbage can be divided into 14 stages (Supplementary Fig. 1a). No obvious differences were found between wild-type (WT) 01-20 and DGMS01-20 plants before the tetrad stage (Fig. 1c and Supplementary Fig. 1a). In the late-tetrad stage (late stage 7), DGMS01-20 plants showed delayed tapetal degeneration without a DNA fragmentation signal, as indicated by terminal transferase-mediated dUTP nick-end labeling (TUNEL) assays (Fig. 1d). After stage 7, in 01-20, microspores were released from the tetrads through degeneration of the pollen mother cell wall and the callose wall; however, in DGMS01-20, the pollen mother cell wall failed to degenerate, and microspores became arrested in tetrads and could not be separated throughout the later developmental stages (Fig. 1c–f and Supplementary Fig. 1a). At stage 9, DGMS01-20 microspores began to degenerate, as indicated by the DNA fragmentation signal, followed by cytoplasm vacuolation, cytoplasm degeneration and collapse (Fig. 1d and Supplementary Fig. 1a). Severe defects in exines and intines were observed: nexines and intines were completely absent; sexines formed but were unevenly deposited, with indistinguishable tecta and bacula in some regions; and pollen coats were produced but could not be deposited on the surface of microspores (Fig. 1e, f). Sexines were highly reduced in later stages (Fig. 1e, f). An abnormal pollen wall structure was also observed by scanning electron microscopy (SEM) (Supplementary Fig. 1b). At stages 13–14, DGMS01-20 anthers

underwent normal dehiscence, but the microspores were morphologically degenerated (Supplementary Fig. 1a).

DGMS01-20 plants were completely identical to 01-20 plants except for pollen development (Supplementary Fig. 2). Female fertility was normal, and the mutant could be maintained by crossing with WT plants, resulting in progeny segregating in accordance with a 1:1 (fertile:sterile) ratio (Supplementary Table 1).

### Map-based cloning of *Ms-cd1*

To determine the causal gene of *Ms-cd1*, we employed a positional cloning approach involving three advanced backcross populations, PO1 (resulting from DGMS01-20 × 01-20; $BC_{24}$), PO2 (resulting from DGMS87-534 × 87-534; $BC_{23}$) and PO3 (resulting from DGMS18k × 18k; $BC_{20}$), which were developed with 79-399-3 as a donor and with backcrossing to the recurrent parents for more than 20 generations (Fig. 2a). PO1 was used for bulked-segregant analysis sequencing (BSA-seq), and a candidate genomic region was detected within 28–31 Mb on C09 ("02-12" reference genome, version 1) (Supplementary Fig. 3). Polymorphic markers were developed to genotype individuals of PO1 and PO2, and *Ms-cd1* was delimited to a 725-kb region. Using the segregating PO3 population (12608 individuals), we ultimately mapped *Ms-cd1* to a 10.9-kb interval (Fig. 2b).

Annotations according to the "02-12" cabbage reference genome in the BRAD database (http://brassicadb.cn) revealed that the 10.9-kb region harbors the promoter region, the first exon and part of the first intron of *Bol035718*. However, no nucleotide variation was present in the coding region of *Bol035718* between the DGMS and WT plants. After sequencing the whole 10.9-kb genomic sequence, we detected only one mutation in the DGMS plants, a 1-bp deletion located 597 bp upstream of the start codon of *Bol035718* (Fig. 2b). Therefore, we speculated that this 1-bp deletion may be responsible for dominant male sterility.

### The expression of *Bol035718* driven by the mutant promoter of DGMS plants results in dominant male sterility

To verify the function of *Bol035718*, a construct containing a 6,028-bp genomic fragment including the mutated promoter region (2913 bp), the *Bol035718* gene (2775 bp) and the terminator region (340 bp) was generated and introduced into cabbage inbred line 01-20 by *Agrobacterium*-mediated transformation (Supplementary Fig. 4). Seven independent transgenic lines showed complete male sterility, identical to that of DGMS01-20, with normal-sized anthers, no viable pollen, microspores arrested in tetrads, an absence of pollen nexines and intines, and complete abortion of microspores that were completely aborted before stage 13 (Fig. 3a–c and Supplementary Fig. 5). Progeny derived from the crossing of male-sterile transgenic plants with WT 01-20 segregated in accordance with a 1:1 ratio (Fig. 3b and Supplementary Table 2). Therefore, we concluded that *Bol035718* corresponds to *Ms-cd1* controlling male sterility and that the 1-bp deletion upstream of *Bol035718* is the causal mutation. Hereafter, the allele causing DGMS is referred to as *Ms-cd1*$_{P\Delta-597}$, and the WT allele is referred to as *Ms-cd1*$_{PWT}$.

### Knocking out *Ms-cd1* results in recessive male sterility different from that of the DGMS type

*Ms-cd1* encodes a transcription factor with a PHD-finger motif that is homologous to MS1 in *Arabidopsis*, PTC1/OsMS1 in rice, and ZmMs7 in maize (Supplementary Fig. 6a)[6,19–22]. Although these homologs are reported to be involved in anther and male gamete development, mutants of the corresponding genes show recessive inheritance and cause completely different male-sterile phenotypes from those of DGMS plants observed in this study. As the only case of a natural allele with a gain of function among *MS1* homologs, *Ms-cd1*$_{P\Delta-597}$ in *B. oleracea* seems to play a role in a different molecular mechanism regulating male sterility.

To further verify the function of *Ms-cd1*, we generated a clustered, regularly interspaced, short palindromic repeats (CRISPR)/CRISPR-

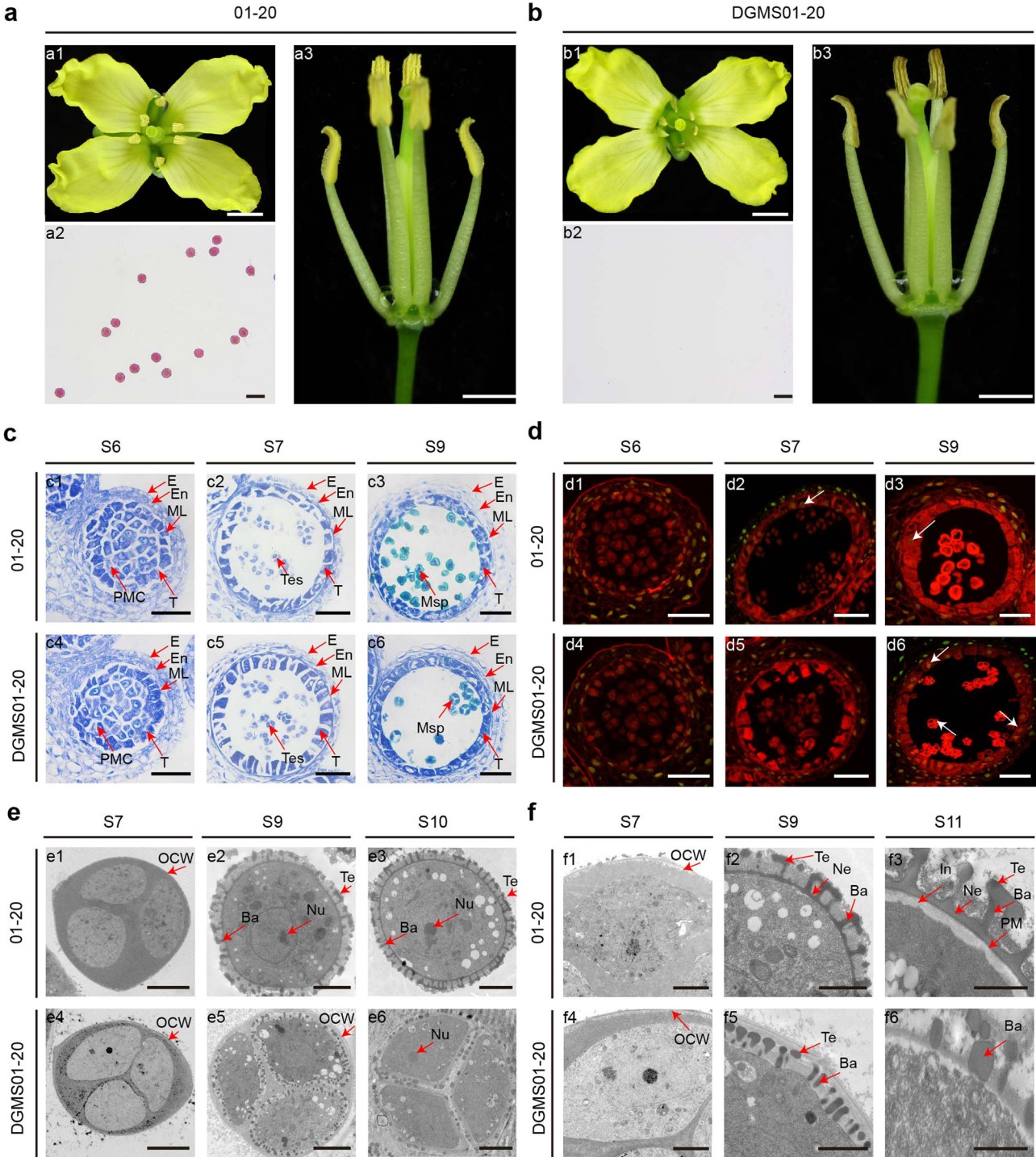

**Fig. 1 | Phenotypes and cytological characteristics of wild-type 01-20 and the DGMS01-20 mutant. a, b** Comparison of flowers (**a1**, **b1**), anthers (**a3**, **b3**) and pollen grains stained with Alexander solution (**a2**, **b2**) among wild-type 01-20 and DGMS01-20 mutants. Scale bar, 3 mm in (**a1** and **b1**), 2 mm in (**a3** and **b3**), 50 μm in (**a2** and **b2**). **c** Transverse section analysis of anthers in the WT and DGMS01-20 mutant at S6 (stage 6), S7 (stage 7) and S9 (stage 9). E epidermis, En endothecium, ML middle layer, T tapetum, PMC pollen mother cell, Tes tetrads, Msp microspore. Scale bar, 50 μm. **d** TUNEL signals of anthers in the WT and DGMS01-20 mutant at

stages S6, S7 and S9. White arrows indicate the TUNEL signals. Scale bar, 50 μm. **e** TEM analysis of microspores in the WT and DGMS01-20 mutant at anther stages S7, S9 and S10. Te tectum, Ba bacula, Nu nucleus, OCW outer cell wall. Scale bar, 4 μm. **f** TEM analysis of the pollen wall in the WT and DGMS01-20 mutant at anther stages S7, S9 and S11. PM microspore plasma membrane, In intine, Ne nexine. Scale bar, 2 μm. Experiments were repeated three times independently with similar results.

associated 9 (Cas9) (CRISPR/Cas9) construct in which the guide RNA targeted the third exon of *Ms-cd1* (Fig. 3a), and we transformed WT 01-20 plants with this construct. Homozygous *ms-cd1PWT* mutants (*ms-cd1PWT*/*ms-cd1PWT*) were completely male sterile but with differences from the DGMS mutant, as the former produced shriveled anthers that

failed to produce pollen grains at the mature stage (Fig. 3a). The anthers and microspores of *ms-cd1PWT* plants developed normally up to stage 7; after this stage, the pollen mother cell wall degenerated normally, enabling the release of microspores. These microspores formed abnormal walls and presented irregular spike-shaped bacula

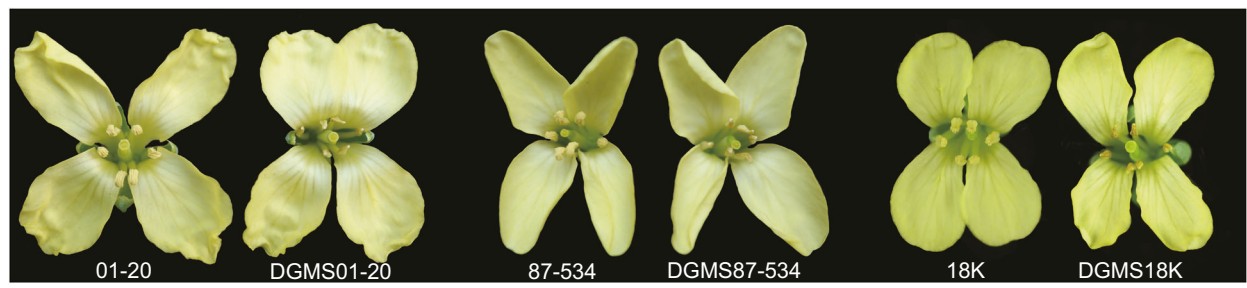

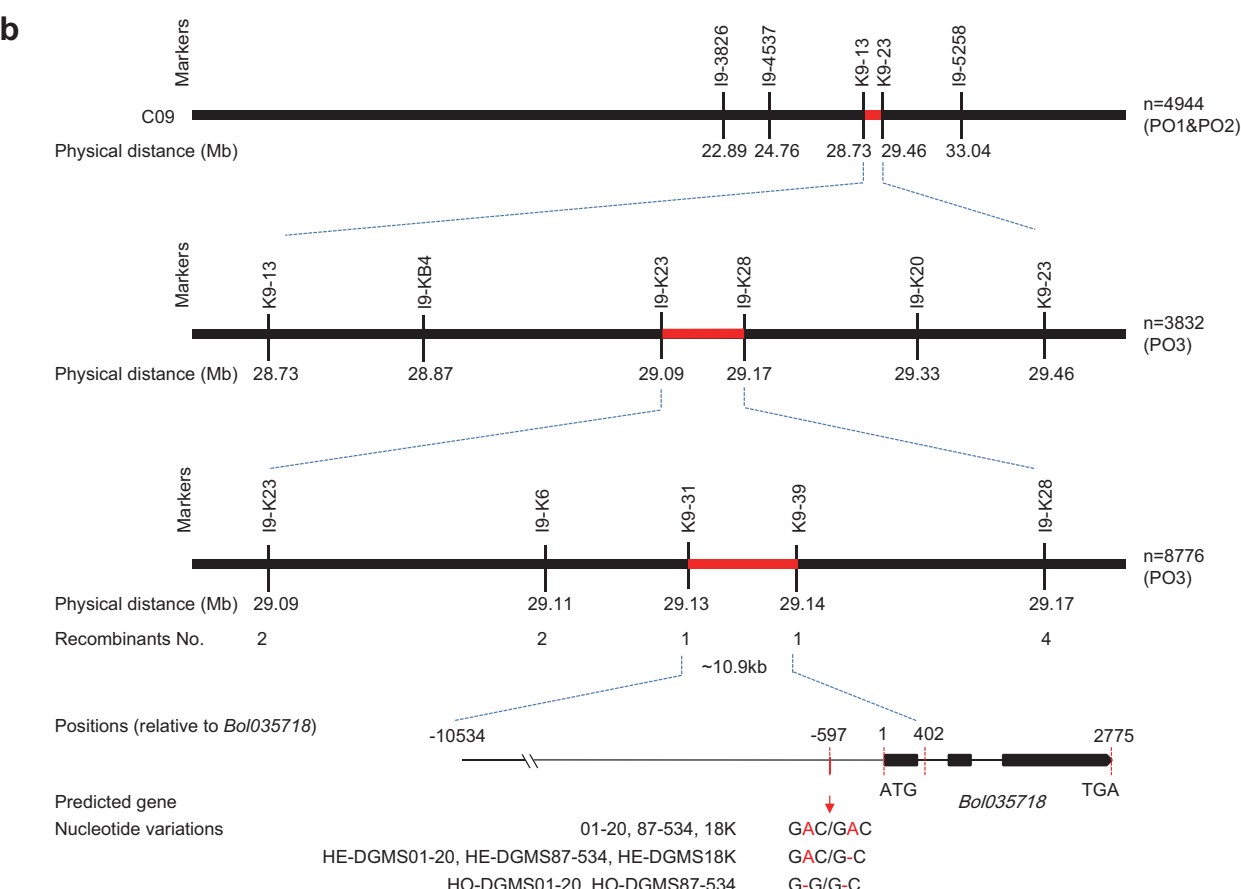

**Fig. 2 | Positional cloning of *Ms-cd1* on chromosome 9. a** Flowers from the parents of the three mapping populations. **b** *Ms-cd1* was mapped to a ~10.9-kb interval between markers K9-31 and K9-39, and allelic variations in this genomic DNA region were evaluated.

(rather than the rod-like bacula of the WT), thinner nexine and intine walls, and a reduced pollen coat. Moreover, microspore cytoplasm vacuolation and degeneration began soon after the release of microspores from tetrads, and complete microspore degeneration resulted in empty locules (Fig. 3d and Supplementary Figs. 5, 7 and 8). Moreover, there were no differences between the *ms-cd1*$_{PWT}$ mutants and WT 01-20 plants in terms of other agronomic traits. F$_1$ plants derived from *ms-cd1*$_{PWT}$ × 01-20 were completely male fertile, and the phenotypes of the F$_2$ progeny resulting from selfed F$_1$ plants fit a 3:1 (fertile:sterile) ratio (Fig. 3d and Supplementary Table 2), indicating that *ms-cd1*$_{PWT}$ is a recessive male-sterile mutant.

We further knocked out *Ms-cd1* in the HE-DGMS01-20 background (*Ms-cd1*$_{PWT}$/*Ms-cd1*$_{PΔ-597}$). After crossing the mutated T0 plants with 01-20 followed by selfing, we obtained *ms-cd1*$_{PΔ-597}$ mutants (*ms-cd1*$_{PΔ-597}$/*ms-cd1*$_{PΔ-597}$). Interestingly, the *ms-cd1*$_{PΔ-597}$ mutant was completely identical to the *ms-cd1*$_{PWT}$ mutant in terms of all examined phenotypic characteristics, including the inheritance of the recessive male sterility trait (Fig. 3a, e, Supplementary Figs. 5, 7 and 8 and Supplementary Table 2).

In summary, we concluded that (I) *Ms-cd1* is required for anther and microspore development; (II) both specific promoter mutation and functional *Ms-cd1* are required for *Ms-cd1*$_{PΔ-597}$ to trigger dominant male sterility; and (III) 79-399-3 DGMS is a gain-of-function male-sterile type whose sterility is distinguishable from that of *ms-cd1* loss-of-function mutants.

## The expression level, but not the tissue-specific expression pattern, of *Ms-cd1* is altered in DGMS plants

Ms-cd1 is localized in the nucleus (Supplementary Fig. 6b). Then, we investigated the spatial expression of the *Ms-cd1* gene via qRT–PCR and

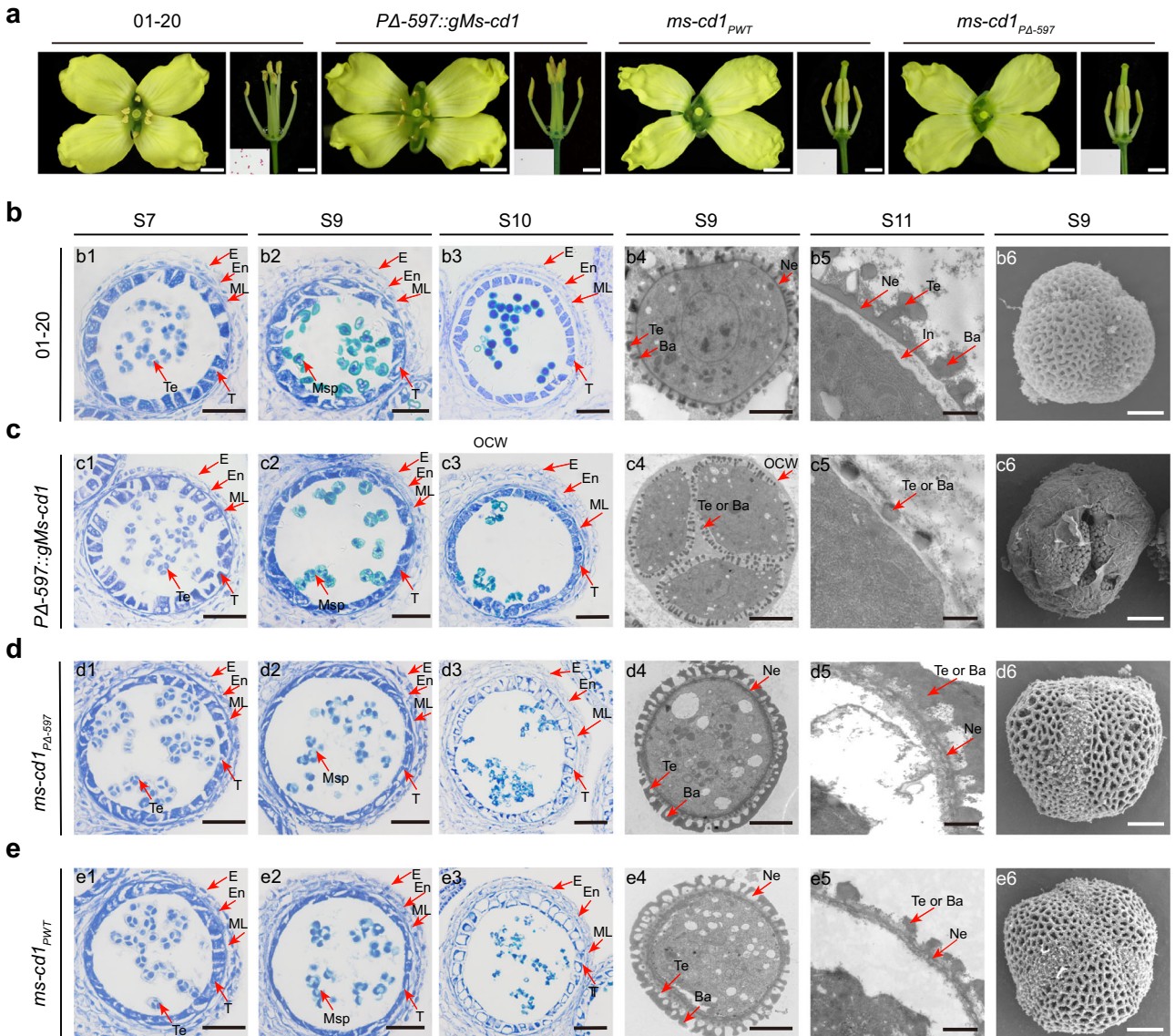

**Fig. 3 | Functional characterization of *Ms-cd1*. a** Flowers, anthers and pollen grains of WT, *PΔ-597::gMs-cd1* and the *ms-cd1_PWT* and *ms-cd1_PΔ-597* mutant plants stained with Alexander solution. Scale bar, 3 mm. **b–e** Transverse section, TEM, and SEM analysis of anthers in WT, *PΔ-597::gMs-cd1* and the *ms-cd1_PWT* and *ms-cd1_PΔ-597* mutant plants. E epidermis, En endothecium, ML middle layer, T tapetum, PMC pollen mother cell, Tes tetrads, Msp microspore, Te tectum, Ba bacula, In intine, Ne nexine, OCW outer cell wall. Scale bar, 50 μm in (**e1–e3**), (**f1–f3**), (**g1–g3**) and (**h1–h3**); 4 μm in (**e4**), (**e6**), (**f4**), (**f6**), (**g4**), (**g6**), (**h4**) and (**h6**); and 1 μm in (**e5**), (**f5**), (**g5**) and (**h5**). Experiments were repeated three times independently with similar results.

in situ hybridization (ISH) assays, which revealed that *Ms-cd1* was specifically expressed in tapetal cells and microspores of anthers at stages 7–8 in both the WT and dominant genic male-sterile plants (Fig. 4a, b). qRT–PCR assays showed that *Ms-cd1* was significantly downregulated in DGMS01-20 compared with 01-20 (Fig. 4b).

We further investigated promoter activities via the analysis of transgenic *Arabidopsis* lines expressing the *GUS* reporter gene under the control of the mutated promoter (*PΔ-597*) and WT promoter (*PWT*). GUS histochemical staining and *GUS* gene expression levels were determined in at least ten independent *PΔ-597::GUS* and *PWT::GUS* transgenic lines, respectively. Intriguingly, in contrast to the qRT–PCR results, *PΔ-597::GUS* showed dramatically enhanced GUS activity and *GUS* gene expression compared with *PWT::GUS*, indicating that *PΔ-597* shows higher promoter activity (Fig. 4b–f). To further verify this result, *ms-cd1_PΔ-597* and *ms-cd1_PWT* mutants were selected to investigate *ms-cd1* expression. The results showed that the expression levels of

*ms-cd1* in *ms-cd1_PΔ-597* were approximately 3-fold higher than those in *ms-cd1_PWT*, which further suggested that PΔ-597 presented much higher promoter activity than PWT (Fig. 4g). Additionally, *ms-cd1* (or *Ms-cd1*) expression in *ms-cd1_PWT* and *ms-cd1_PΔ-597* was dramatically enhanced compared with that in *Ms-cd1_PWT* and *Ms-cd1_PΔ-597* plants, suggesting that functional *Ms-cd1* repressed its own expression (Fig. 4g). Considering the observed self-repression and short-stage expression characteristics, we speculated that the transcriptional ability of *Ms-cd1* could not be accurately measured by qRT–PCR.

## BoERF1L regulates male fertility via the direct repression of *Ms-cd1_PWT* expression

To investigate why the promoter mutation affected the expression of *Ms-cd1*, we scanned the promoter sequence with the online tool PLACE (http://www.dna.affrc.go.jp/PLACE/signalscan.html), but no known *cis*-element was identified near the -597 bp region of the promoter.

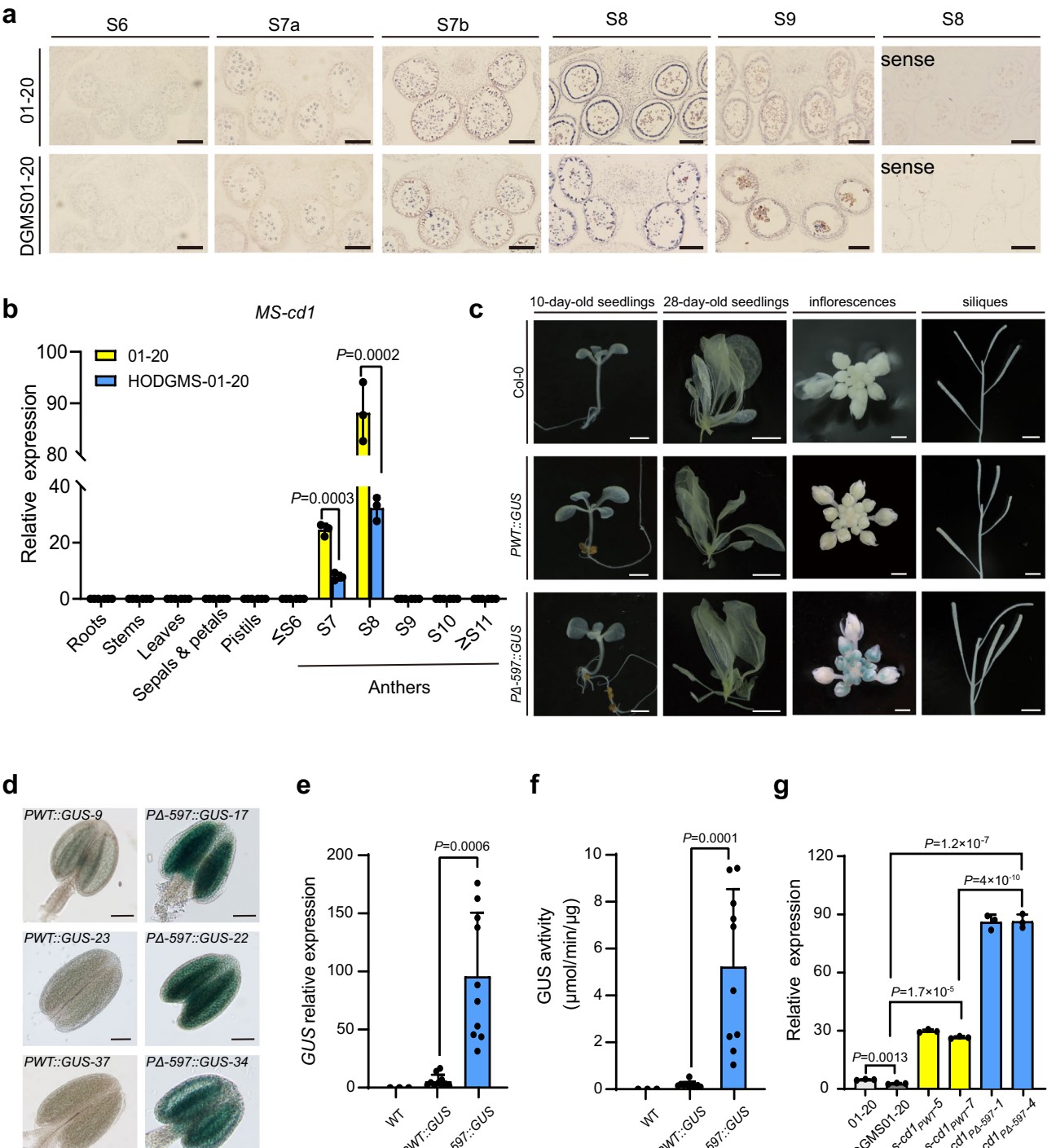

**Fig. 4 | Expression pattern and promoter activity of *Ms-cd1*. a** In situ hybridization of *Ms-cd1* mRNA in anthers of WT and DGMS01-20 plants at anther stages S6–S9. Hybridization with sense probes was negative in S8. Scale bar, 100 μm. **b** qRT–PCR analysis of *Ms-cd1* expression in various tissues of 01-20 and DGMS01-20. **c, d** GUS staining of seedlings, inflorescences, siliques and anthers from representative *PWT::GUS* and *PΔ-597::GUS Arabidopsis* plants. Scale bars, 2 mm for 10-day-old seedlings, 1 cm for 28-day-old seedlings, 1 mm for inflorescences, 5 mm for siliques and 25 μm for anthers. **e** GUS gene expression in *PWT::GUS* and *PΔ-597::GUS*. Each dot indicates an independent transgenic line. **f** GUS activity in *PWT::GUS* and *PΔ-597::GUS*. Each dot indicates an independent transgenic line. **g** *Ms-cd1* (or *ms-cd1*) expression levels in WT 01-20, DGSM01-20, *ms-cd1_{PWT}* and *ms-cd1_{PΔ-597}*. Data are presented as the means ± SD (*n* = 3 for (**b**) and (**g**), *n* = 10 for (**e**) and (**f**)). A two-tailed unpaired *t*-test was used for statistical analysis. Experiments were repeated three times independently with similar results. Source data are provided as a Source Data file.

To identify the upstream regulators of *Ms-cd1*, we constructed a cabbage complementary DNA (cDNA) library and applied a yeast one-hybrid (Y1H) approach to search for proteins associated with the *Ms-cd1* promoter. A 66-bp WT promoter fragment (624–559 bp upstream of the *Ms-cd1* start codon) was used as bait to screen the cabbage cDNA library. Most of the positive clones (listed in Supplementary Table 3) obtained were derived from the same *BolO28757* gene, which encodes a *B.*

*oleracea* ethylene response factor 1-like (BoERF1L) transcription factor (Supplementary Fig. 9).

To confirm the interaction between BoERF1L and the bait, we fused the full-length coding sequence of BoERF1L to the yeast GAL4 transcriptional activation domain (GAL4-AD). *Ms-cd1* promoter fragments (-624 to -559) WTpro (from *Ms-cd1_PWT*) and MTpro (from *Ms-cd1_PΔ-597*) (Supplementary Fig. 9) were inserted into the pAbAi vector as reporters. When pBoERF1L-AD was cotransformed into Y1H gold yeast cells, it significantly activated aureobasidin A (AbA) resistance in WTpro-pABAi but not MTpro-pABAi (Fig. 5a). To confirm the binding ability, we subsequently performed an electrophoretic mobility shift assay (EMSA). The assay results showed that BoERF1L-His could directly bind WTpro but not MTpro in vitro. The unlabeled WTpro probe competed with the corresponding labeled probe and reduced the abundance of shifted bands in a concentration-dependent manner (Fig. 5b), confirming the binding specificity of the interaction. We further divided WTpro into seven 16-bp segments, and the results showed that BoERF1L-His bound to W2-W6 but not W1 and W7 and had a higher affinity for W4 than for W2, W3, W5, and W6 (Supplementary Fig. 10). These data support a direct interaction between BoERF1L and the promoter of *Ms-cd1_PWT* from −604 to −589, with GAC as the core binding sequence, however, in the *Ms-cd1_PΔ-597* promoter, the 1-bp deletion prevented this binding ability. Finally, a quantitative dual-luciferase transactivation assay (dual-LUC) was conducted to confirm the effects of BoERF1L on *Ms-cd1_PWT* transcription. A constitutively expressed *Renilla luciferase* gene driven by the 35S promoter was used as an internal reference reporter. The dual-LUC assay results showed that the BoERF1L effector strongly repressed *PWT::LUC* reporter transcription, whereas the negative control did not (Fig. 5c). Taken together, these results indicated that BoERF1L directly binds to the specific site of the *Ms-cd1_PWT* promoter and represses *Ms-cd1_PWT* expression.

*BoERF1L* is highly expressed in anthers before stage 9 (Fig. 5d). To determine whether *BoERF1L* plays a role in anther and microspore development, we knocked out *BoERF1L* by the CRISPR/Cas9 method in the 01-20 background (Supplementary Fig. 11a, b). The *boerf1* mutants showed significantly reduced male fertility, with approximately 50% pollen viability (Fig. 5e, f and Supplementary Fig. 11c). Transmission electron microscopy (TEM) and SEM assays showed that, as in the DGMS lines, many microspores could not produce intine walls (Fig. 5g). We further knocked down *BoERF1L* via RNA interference (RNAi) in the 01-20 background. Two representative lines (RNAi-4 and RNAi-6) were selected from eight independent lines for detailed studies (Supplementary Fig. 12a–c). These lines showed a similar phenotype to the *boerf1* mutants, with pollen viability of 61.2% for RNAi-4 and 56.6% for RNAi-6 (Supplementary Fig. 12d, e). In addition, we assessed the expression of *Ms-cd1* and its closely related genes in *boerf1*, which showed that all the evaluated genes presented expression level alterations identical to those in the DGMS lines (Supplementary Figs. 11d and 13). These results suggested that *BoERF1L* affected the male fertility of *B. oleracea* via the regulation of *Ms-cd1*.

### *Ms-cd1_PΔ-597* and *Ms-cd1* loss-of-function alters the expression of genes required for tapetum and pollen development

As *MS1* (the ortholog of *Ms-cd1*) is reported to be involved in the *DYT1-TDF1-AMS-MS188-MS1* network, which is critical for anther and pollen development[23,24], we investigated the expression of these genes and their downstream targets. Most of them showed contrary expression levels in DGMS01-20 and *ms-cd1* (Supplementary Fig. 13). The upstream genes *BoTDF1*, *BoAMS* and *BoMS188* were downregulated in DGMS01-20, but upregulated in *ms-cd1*, indicating that *Ms-cd1* played a feedback repression role in the *BoDYT1-BoTDF1-BoAMS-BoMS188-Ms-cd1* module. The *BoQRTs* genes, required for tetrad wall degeneration, and the *BoTEK* gene, required for pollen wall assembly were dramatically downregulated in DGMS01-20, consistent with the defects of microspore arrest in tetrads, identical to what occurs in the *qrt*

mutant[25], and absent nexine and intine layers, similar to what occurs in the *tek* mutant[26] (Fig. 1 and Supplementary Fig. 13a). Other pollen wall development requiring genes, such as *BoYP704B1*, *BoPKSA*, and *BoPKSB* were also downregulated in DGMS01-20 and upregulated in *ms-cd1*. Interestingly, pollen coat-related genes such as *BoLTPs*, *BoEXLs* and *BoGRPs* were upregulated in DGMS01-20 and downregulated in *ms-cd1*, indicating that *Ms-cd1* may be an activator of pollen coat development (Supplementary Fig. 13a).

Based on the above results, a working model of the regulation of male fertility by *Ms-cd1* is proposed (Supplementary Fig. 13b). *Ms-cd1* is directly targeted by the transcription factor BoERF1L to precisely maintain a stable expression level, which is essential for the feedback regulation of the *BoDYT1-BoTDF1-BoAMS-BoMS188-Ms-cd1* network and a subsequent series of genes required for pollen development. In DGMS, promoter mutation causes dysregulation of *Ms-cd1* and strong negative feedback of the *BoDYT1-BoTDF-BoAMS-BoMS188-Ms-cd1* module and, subsequently, pollen wall development-related genes such as *BoQRTs*, *BoTEK* and *BoPKSA*. In RGMS, *ms-cd1* cannot feed back into the module, which results in the upregulation of some genes, such as *BoTEK* and *BoPKSA*, and the downregulation of other genes related to pollen development, such as *BoLTPs* and *BoEXLs*. Overall, our results suggest that the precise regulation of *Ms-cd1* is vital for male fertility.

### A DGMS system for heterosis utilization in dicotyledonous and monocotyledonous crop species

To further test whether *Ms-cd1_PΔ-597* can induce male sterility in other plant species, we introduced *PΔ-597::Ms-cd1* into *Arabidopsis* ecotype Columbia (Col-0), rapeseed (Westar), tomato (Ailsa Craig, AC) and rice (ZH11). More than five transgenic lines were generated for each plant species; these *PΔ-597::Ms-cd1* lines exhibited normal vegetative morphology and female fertility identical to those of their corresponding WT plants but showed stable male sterility, produced no viable pollen and failed to produce seeds upon selfing (Fig. 6a–d). Progenies derived from a cross between *PΔ-597::Ms-cd1* and WT plants segregated at a 1:1 (fertile:sterile) ratio (Supplementary Table 4). The results indicated that ectopic expression of *Ms-cd1_PΔ-597* induced dominant male sterility and was conserved in dicotyledonous and monocotyledonous plants.

Exploiting male sterility to increase crop yields is feasible and vital globally. In contrast to other male sterility systems, the DGMS system has greater value, with the advantages of increasing crop yields, good efficiency and safety to the ecosystem[4]. The currently applied *Ms-cd1_PΔ-597* DGMS system poses disadvantages including difficulties in generating, propagating and preserving HO-DGMS lines. Herein, we propose the following five steps for using the DGMS system. The detailed strategy is shown in Fig. 6e, with cabbage as an example.

Step 1: generation of HE-DGMS lines. The DGMS gene is introduced from a donor plant to an elite line by backcrossing for 4−5 generations, or a DGMS mutant is directly generated via precise genome editing of the *Ms-cd1_PΔ-597* promoter and elimination of transgene elements in Step 2.

Step 2: generation and propagation of HO-DGMS lines by in vivo doubled haploid (DH)/haploid induction. First, HE-DGMS lines are pollinated with pollen from a DH/haploid inducer, such as the DH inducer Y3380[27] and *dmp*-based haploid inducer[10]. Among the progeny, DHs/haploids can be identified by phenotypic markers, molecular markers or fluorescent markers. When using haploid inducers, haploids must be converted to DH plants via colchicine (or other methods). For the propagation of HO-DGMS lines, the cross-pollination of HO-DGMS and DH/haploid inducer lines can be performed by using pollinators in protected fields.

Step 3: large-scale reproduction of a 100% HE-DGMS line. By crossing HO-DGMS with its corresponding maintainer line, breeders can obtain HE-DGMS lines with 100% male-sterile plants. This DGMS system requires only a few HO-DGMS plants, which can be easily

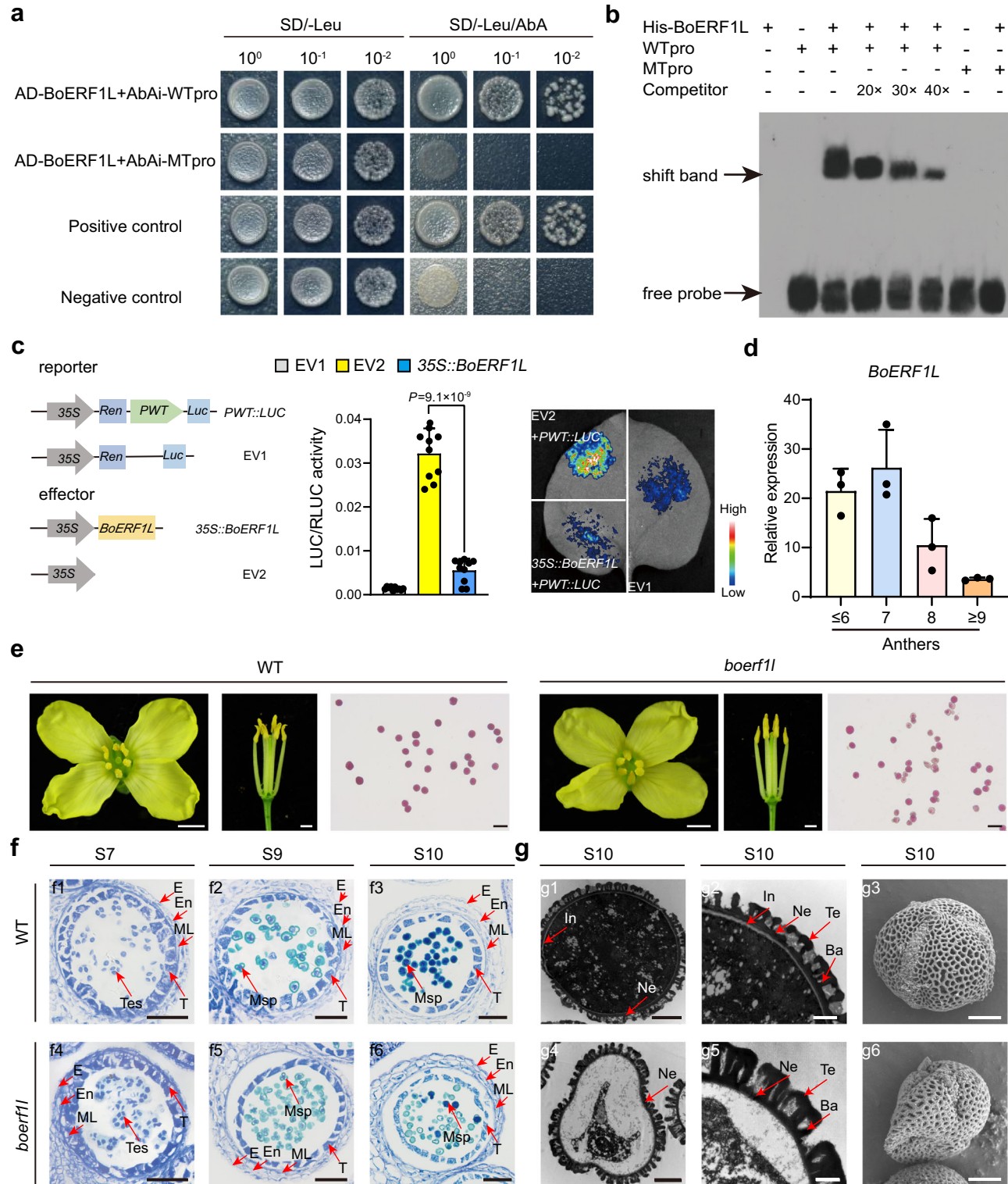

**Fig. 5 | BoERF1L regulates male fertility by the direct repression of *Ms-cd1* expression. a** Yeast one-hybrid assay. WTpro represents a 66-bp probe (−624 to −559 upstream of ATG) from the WT promoter. MTpro represents a 65-bp probe (−624 to −559 upstream of ATG) from the DGMS01-20 promoter. Yeast strains were cultured on SD/-Leu medium with 200 ng/mL AbA. **b** EMSA analysis. The biotin-labeled probes are indicated. Competition for BoERF1L binding was performed with excess unlabeled WTpro probe at 20×, 30× and 40× the amount of labeled probe. **c** The dual-luciferase transient expression assay shows that BoERF1L represses transcription of the *Ms-cd1_PWT* gene promoter. Luciferase intensity was imaged and quantified using a Tanon 5200 system. **d** qRT–PCR analysis of *BoERF1L*

expression in various tissues of 01-20. **e** Flowers, anthers and pollen grains stained with Alexander solution among WT and *boerf1l* mutants. Scale bars, 3 mm for flowers, 2 mm for anthers, 50 μm for pollen grains. **f**, **g** Transverse section, TEM and SEM analysis of anthers in WT and the *boerf1l* mutant. E epidermis, En endothecium, ML middle layer, T tapetum, Tes tetrads, Msp microspore, Te tectum, Ba bacula, In intine, Ne nexine. Scale bars, 50 μm in (**f1**–**f6**); 4 μm in (**g1**), (**g2**), (**g4**) and (**g6**); 2 μm in (**g2**) and (**g5**). Data are presented as the means ± SD (*n* = 10 for (**c**), *n* = 3 for (**d**)). A two-tailed unpaired *t*-test was used for statistical analysis. Experiments were repeated three times independently with similar results. Source data are provided as a Source Data file.

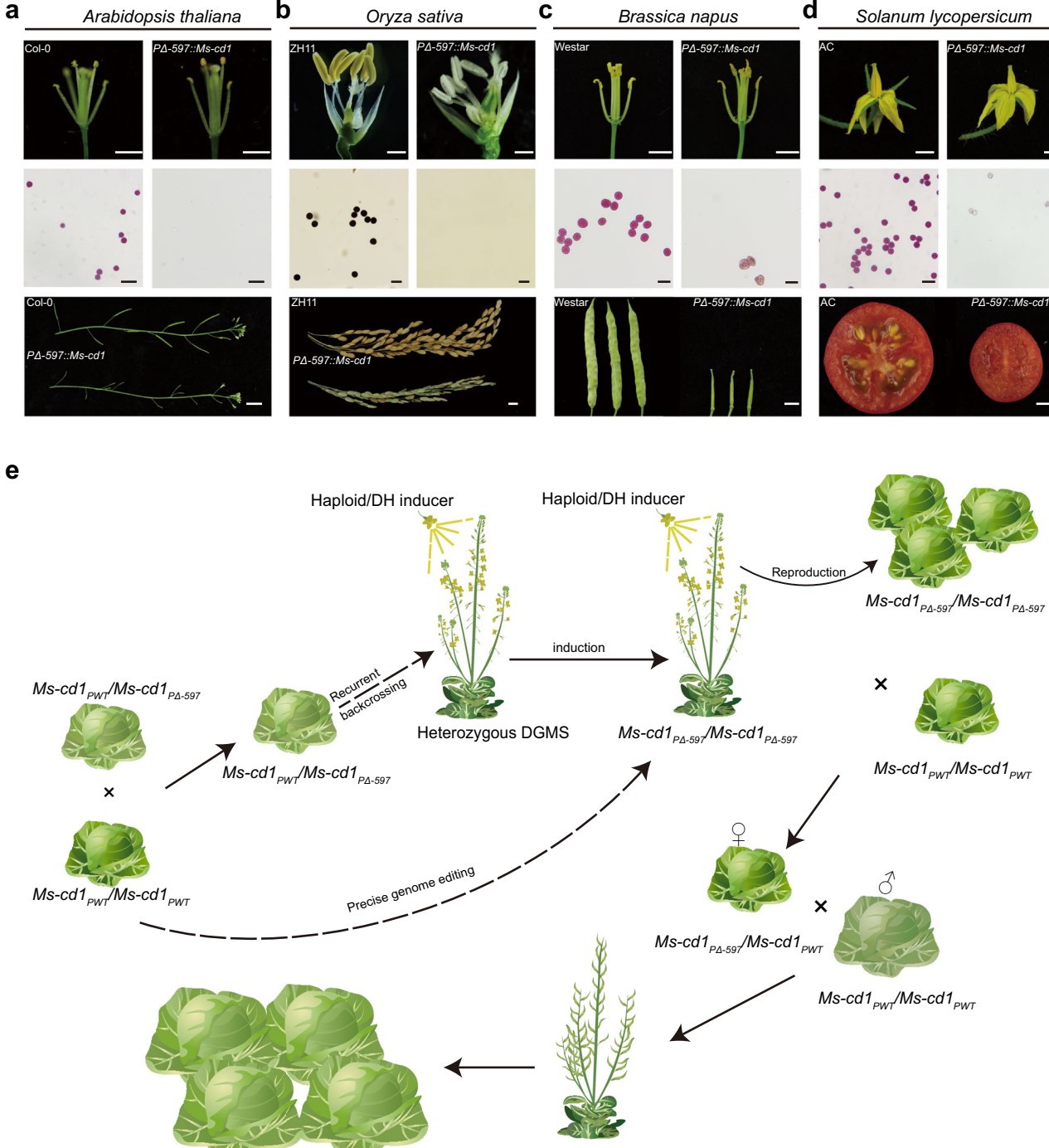

**Fig. 6 | *Ms-cd1$_{PΔ-597}$*-induced dominant male sterility is conserved in dicotyledonous and monocotyledonous plants. a–d** Flowers/anthers (scale bars, 1 mm in (**a**), 4 mm in (**b**), 2 mm in (**c** and **d**)), pollen grains stained with Alexander solution or I2-KI solution (scale bars, 50 μm), and seed setting upon selfing among wild-type and *PΔ-597::Ms-cd1* transgenic lines of *Arabidopsis* Col-0, rice ZH11, rapeseed Westar and tomato Ailsa Craig (AC) (scale bars, 1 cm). **e** Workflow of the DGMS system for heterosis utilization in cabbage. Haploid/DH inducers play a key role in the creation and propagation of HO-DGMS lines. Experiments were repeated three times independently with similar results.

produced by DH/haploid inducer lines with typical haploid induction rates of 1%-10% (taking cabbage as an example, by crossing with the maintainer line and then crossing with the male parent, one HO-DGMS plant would produce 5000 HE-DGMS individuals, and approximately 25 million F₁ seeds, which is sufficient to cultivate an area of 400 ha).

Step 4: hybrid seed production. Hybrid seeds can be obtained by crossing the HE-DGMS line with another elite male parent.

Step 5: crop production. For crops whose vegetative organs are harvested (e.g., cabbage), F₁ hybrids are cultivated and harvested in the same manner as traditional hybrids (Fig. 6e). For crops whose fruits (e.g., tomato) or seeds are harvested (e.g., maize, rice), F₁ hybrids consist of 50% male-fertile plants and 50% male-sterile plants; when both types are planted in the field, the pistils of male-sterile plants can be pollinated by male-fertile plants; this allopollination promotes

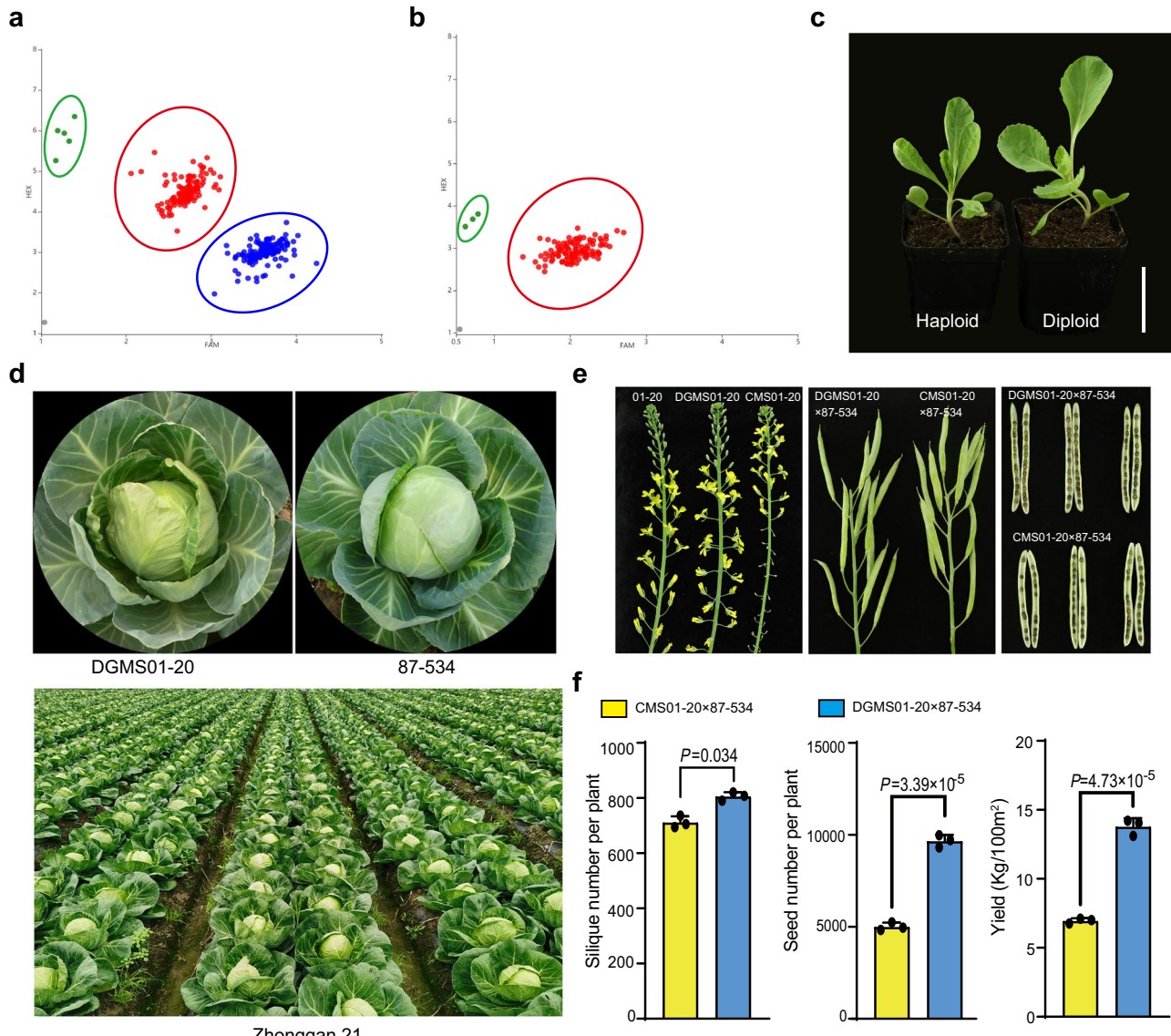

**Fig. 7 | The cabbage hybrid Zhonggan 21 produced via the DGMS-based system and the DGMS-based system enhances hybrid seed yield compared with the commonly used CMS-based system. a** Detection of haploid HO-DGMS from progenies of the HE-DGMS01-20 × *boc03.dmp9* inducer line using KASP marker K9-6. *boc03.dmp9* inducer line, a maternal haploid inducer line created previously[15]. K9-6, a KASP marker developed based on the causal mutation (−597 upstream of the start codon) of *Ms-cd1$_{PA-597}$*. Green dots, homozygous DGMS individuals (i.e., haploid HO-DGMS01-20); red dots, heterozygous DGMS individuals; blue dots, homozygous wild-type individuals; gray dot, negative control. **b** Detection of haploid HO-DGMS from progenies of the HO-DGMS01-20 × *boc03.dmp9* inducer line using the KASP marker K9-6. **c** Phenotypes of haploid and diploid HO-DGMS01-20. Diploid HO-DGMS01-20 plants were produced from haploid HO-DGMS01-20 via colchicine. Scale bars, 5 cm. **d** Female parent DGMS01-20, male parent 87-534 and

hybrid Zhonggan 21. DGMS01-20 is generated by a *dmp*-based inducer. **e** Comparison of inflorescences and seed setting performance of the 01-20, DGMS01-20 and CMS01-20 lines. **f** Silique number, seed number and seed yield of DGMS01-20 and CMS01-20 lines, when crossed to the male parent 87-534. Three field replicates were performed. For each field experiment, 330 female DGMS01-20 and 165 male 87-534 were planted in a 100-m² area. Female plants were pollinated by honey bees (*Apis mellifera ligustica*). Thirty female individuals were randomly selected for the calculation of silique numbers and seed numbers. Seeds were harvested and weighed from all female individuals in each field experiment. Data are presented as the means ± SD (*n* = 3 for (**f**)). A two-tailed unpaired *t*-test was used for statistical analysis. Experiments were repeated three times independently with similar results. Source data are provided as a Source Data file.

increased energy allocation to female organs and thereby increases crop yields[4].

Using cabbage lines HE-DGMS01-20 and HE-DGMS21-3, we generated and propagated HO-DGMS lines using *dmp*-based inducer lines (Fig. 7a–d). When the lines were used for hybrid seed production, we observed that hybrid seed yields increased by more than 98% compared with those of hybrids developed via the traditional CMS-based method, and F₁ cabbage hybrids showed high quality and uniformity (Fig. 7e, f and Supplementary Fig. 14a, b).

Together, the stable *Ms-cd1$_{PA-597}$* gene and its associated DGMS system are applicable for hybrid seed production in crops.

## Discussion

As an effective pollen control tool for cross-pollination and heterosis utilization, male sterility is considered one of the most important traits extensively studied in plants[28,29]. However, only a few genes controlling male sterility have been identified in *B. oleracea*[30]. In particular, the *Ms-cd1* DGMS *B. oleracea* line (79-399-3 mutant) is a valuable male-sterile genetic resource that has been applied to cole vegetable breeding and

seed production in China. However, the *Ms-cd1* gene has not been identified[16]. In this study, using a positional cloning approach, we cloned *Ms-cd1*. We show that *Ms-cd1* encodes a PHD-finger transcription factor, and its orthologs have been identified in several plant species: *MS1* in *Arabidopsis*[19,20], *OsMs1/OsPTC1* in rice[21,31], and *ZmMs7* in maize[6,22]. Loss of function of *Ms-cd1* orthologs results in recessive male sterility, similar to that of the *ms-cd1* mutant in this study, indicating that *Ms-cd1* plays a conserved role in regulating male reproductive development in both dicots and monocots. Among its orthologs, *Ms-cd1$_{P\Delta-597}$* is a unique natural dominant male-sterile allele resulting from a mutation in the noncoding region and conferring a unique mutant phenotype to anthers and microspores. The special promoter mutation and functional coding region of *Ms-cd1$_{P\Delta-597}$* are necessary for the *Ms-cd1* gene to cause male sterility, as the *ms-cd1$_{P\Delta-597}$* loss-of-function allele is identical to *ms-cd1$_{PWT}$*. *Arabidopsis* MS1 functions downstream of MS188 and is involved in the well-known DYT1-TDF1-AMS-MS188-MS1 network, a feed-forward module regulating tapetum and pollen wall development by targeting a series of genes, such as *QRTs*, *EXLs*, *CYP703A2*, *TEK*, and *MS2*[23,24,26,32]. Nevertheless, the precise molecular mechanism of *MS1* in this module remains largely unknown. *Ms-cd1$_{P\Delta-597}$* alters the expression of genes required for tapetum and pollen development, including its upstream genes in the DYT1-TDF1-AMS-MS188-MS1 module, indicating that *Ms-cd1* is involved in negative feedback regulation of this network. Consistent with the downregulation of *BoQRTs* and *BoTEK*, DGMS plants showed some key defects similar to what is observed in the *qrt* mutant (microspores arrested in tetrads)[25,33] and the *tek* mutant (absence of nexine and intine layers of microspores)[26]. It will be interesting to investigate how *Ms-cd1* is involved in the feedback regulation of DYT-TDF-AMS-MS188-*Ms-cd1* in the future.

Ethylene response factors (ERFs) belong to the plant-specific APETALA 2 (AP2) superfamily[34]. As transcription factors that respond to ethylene signals, ERFs play important roles in plant development and the response to both pathogen attack and abiotic stresses through specific binding to GCC-boxes (AGCCGCC) and DRE/CRT core motifs (A/GCCGAC) in the promoter of downstream genes[35,36]. Interestingly, we identified the ERF transcription factor BoERF1L, which directly binds to a motif with a GAC-core sequence in the promoter of *Ms-cd1$_{PWT}$*, as confirmed by the results of our in vivo and in vitro assays. *Arabidopsis* ERF1, an ortholog of BoERF1L, serves as a key integrator of the ethylene and jasmonic acid pathways involved in regulating the defense response and plant development[37,38]. In *B. oleracea*, BoERF1L represses the expression of *Ms-cd1$_{PWT}$* through its ability to bind to the GAC-core motif in the *Ms-cd1$_{PWT}$* promoter. This binding ability is disrupted by the 1-bp deletion in the promoter of *Ms-cd1$_{P\Delta-597}$*, causing dysregulation of *Ms-cd1* and, ultimately, male sterility. In addition, knocking down the *BoERF1L* gene resulted in reduced male fertility, further confirming its role in male gamete development. These results suggest a molecular mechanism through which BoERF1L plays an important role in the precise regulation of *Ms-cd1* and the maintenance of male fertility.

DGMS is an ideal tool for plant breeding and seed production, with advantages including increases in production through all methods of pollination, cost effectiveness without the need for seed sorting, and feasibility of its use in hybrid breeding to pyramid multiple genes[4,7]. To facilitate the use of the DGMS gene *Ms-cd1$_{P\Delta-597}$* in crop plants, we introduced it into other plant species, which resulted in stable dominant male sterility in rice, *Arabidopsis*, tomato, and rapeseed, indicating that *Ms-cd1$_{P\Delta-597}$*-induced dominant male sterility is conserved in both dicotyledonous and monocotyledonous plants. Considering that *Ms-cd1* is evolutionarily conserved in plant species[21,22], DGMS plants can be directly generated by promoter modification in various crop species (especially for those without DGMS mutants) via genome editing technology. Therefore, *Ms-cd1$_{P\Delta-597}$* and its orthologs have great potential for hybrid breeding and seed production in multiple crop species. However, the lack of efficient

systems has limited the utilization of DGMS lines. Currently, two types of DGMS systems have been developed: biotechnology-based male sterility systems (which have been used in several cereal crop species) and the *Ms-cd1$_{P\Delta-597}$* DGMS system (which has been used in cole vegetables)[6,8,15]. Biotechnology-based male sterility systems produce DGMS female lines and hybrids containing transgenic elements, which are regulated and limited in some countries[4]. However, the *Ms-cd1$_{P\Delta-597}$* DGMS system presents two disadvantages: (1) it is difficult to generate HO-DGMS plants, because it relies on the occurrence of trace amounts of pollen grains from HE-DGMS plants, and (2) HO-DGMS plants must be preserved and propagated by tissue culture, which is inconvenient and expensive[16]. By exploiting the in vivo DH/haploid induction methods, we proposed an advanced DGMS system in which an HO-DGMS line can be generated, propagated and preserved by seed. Overall, this system is transgene element free, simple, cost effective, and applicable to multiple crop species. The results of a field test showed that, compared with the CMS system, the DGMS system increased hybrid cabbage seed yield by 98%. We believe that the application of this system can also increase the yield of grain and other crop species whose seeds are harvested as products. In addition, DGMS lines can be used to integrate and pyramid multiple elite traits, such as high yield and resistance to abiotic and biotic stresses, in recurrent selection breeding.

In conclusion, we report the cloning of *Ms-cd1* and elucidate that it is regulated by *BoERF1L* to control male fertility in a dominant manner in cabbage. Based on this, we propose a DGMS system for hybrid seed production. The *Ms-cd1* gene and its associated DGMS system may facilitate hybrid breeding in *B. oleracea* and other major crop species.

## Methods

### Plant materials and growing conditions

We used 10 cabbage (*B. oleracea* var. *capitata*) lines, 3 cabbage advanced backcross populations, 2 cabbage hybrids, 1 *Arabidopsis thaliana* line, 1 rice (*Oryza sativa* L. ssp. *japonica* "ZH11") line, 1 tomato (*Solanum lycopersicum* "Ailsa Craig") line, 1 rapeseed (*B. napus* "Westar") line, and 1 *Nicotiana benthamiana* line in this study. These plant materials and transgenic lines and the gene editing lines generated from them are listed in Supplementary Data 1. These plant materials were grown at the Institute of Vegetables and Flowers, Chinese Academy of Agricultural Sciences, in Beijing. All cabbage plants were grown in open fields from August to mid-October and then transplanted to greenhouses. After vernalization under cold conditions (0–10 °C), they flowered in April of the following year. All rice, tomato, and rapeseed plants were grown and maintained regularly in the experimental field under local growing conditions in Beijing, China. *A. thaliana* Col-0 and *N. benthamiana* were planted in the soil and maintained in growth chambers under long-day conditions (16 h light, 8 h dark) at 22 °C. Transgenic *Arabidopsis* plants were screened via 1/2-strength Murashige and Skoog media supplemented with 30 mg/L hygromycin B. The hygromycin-resistant seedlings were transplanted to soil and grown under the same conditions as the WT plants.

### Phenotypic characterization of plant materials

Images of whole plant, flower, silique, inflorescence and fruit morphology were captured with a Canon EOS M6 digital camera (Canon, Tokyo, Japan). Rice and *Arabidopsis* anthers were imaged with a SZX2-ILLB16 stereomicroscope (Olympus, Tokyo, Japan).

To evaluate pollen viability, anthers of three representative individuals were analyzed. Pollen grains from dicotyledonous species, including cabbage, tomato, rapeseed, and *A. thaliana*, were stained with Alexander's solution, and grains from rice plants were stained with 1% I2-KI solution and imaged with a BX-53TR microscope (Olympus, Tokyo, Japan). Darkly stained, round pollen grains were defined as viable, whereas unstained, lightly stained and irregular pollen grains were defined as sterile.

For transverse section analysis, fresh anthers of *B. oleracea* at different developmental stages were sampled, fixed in formalin–acetic acid–alcohol (FAA) solution (a mixture of 37% formaldehyde, 70% ethanol, and 100% acetic acid), dehydrated through an ethanol series (70%, 85%, 95%, and 100%), subjected to transverse sectioning, embedded and cut into 2-µm sections using a Leica RM2265 microtome (Leica, Germany). The sections were further stained with 0.05% toluidine blue O and observed under a BX-51TRF microscope (Olympus, Tokyo, Japan).

For SEM analysis, the anthers were fixed and dehydrated through an ethanol series (50%, 70%, 80%, 90%, and 100%), using the critical-point drying method, and then coated with gold and imaged using a scanning electron microscope (HITACHI HT7700, Japan).

For TEM analysis, anthers were washed in 0.1 M phosphate buffer and embedded in epoxy resin. Ultrathin sections (50–70 nm) were cut using a Leica EM UC7 ultramicrotome (Leica Microsystems, Belgium) and stained with uranyl acetate and lead citrate. Sections were imaged using a transmission electron microscope (HITACHI, Regulus 8100, Japan).

For TUNEL assays, anthers at different developmental stages were collected from WT 01-20 and DGMS01-20 plants. DNA fragmentation was detected via TUNEL assays using a TUNEL kit (G1502, Servicebio) according to the manufacturer's instructions. The samples were subsequently evaluated under a fluorescence microscope (Nikon Eclipse C1; Nikon, Tokyo, Japan).

## Map-based cloning of *Ms-cd1*

The male-sterile lines HE-DGMS01-20, HE-DGMS87-534, and HE-DGMS18K were used as female parents in crosses with their corresponding maintainer lines to generate three advanced backcross populations: PO1 (resulting from HE-DGMS01-20 × 01-20; BC$_{24}$), PO2 (resulting from HE-DGMS87-534 × 87-534; BC$_{23}$) and PO3 (resulting from HE-DGMS18k × 18k; BC$_{20}$). Thirty male fertile and thirty male sterile plants were randomly selected from the PO1 population to construct DNA pools. The bulked DNA samples were extracted using a Plant Genomic DNA Kit (Tiangen, Beijing, China), and then sequenced at the Beijing Genomics Institute (BGI) (Beijing, China), with an Illumina Hi-Seq 2500 platform. The resulting data were filtered and aligned to the cabbage reference genome ("02-12" reference genome, version 1). SNP- index and Δ(SNP- index) were calculated. Following the standard protocol for BSA-based mapping, *Ms-cd1* was primarily delimited to a genomic interval on C09. A total of 13 markers within the interval were designed using Primer 3 to genotype the plants among the populations. *Ms-cd1* was ultimately delimited to a 10.9-kb region.

To determine the nucleotide variations between the male-fertile and male-sterile lines, the entire genomic regions were amplified and sequenced from 01-20, HE-DGMS01-20, HO-DGMS01-20, 87-534, HE-DGMS HO-DGMS87-534, 18K and HE-DGMS18K. A list of primers for positional cloning and DNA fragment amplification is shown in Supplementary Data 2.

## Plasmid construction and *Agrobacterium*-mediated plant genetic transformation

For functional complementation, a fragment 6028 bp in length containing the mutated promoter region, *Ms-cd1* gene coding region and terminator region (340 bp) was amplified from DGMS01-20 genomic DNA and inserted into a modified pCAMBIA1301 vector to generate *PΔ-597::gMs-cd1*.

To knock out the *Ms-cd1* gene, a single guide RNA (sgRNA) sequence (5′-ACGATGCATCGCACAAGAAAGGG-3′) targeting the *Ms-cd1* coding region was designed and inserted into a modified pKSE401 CRISPR/Cas9 plasmid to generate *Ms-cd1*-CRISPR/Ca9. To knock out the *BoERF1L* gene, a sgRNA sequence (5′-ATTATGTA-CAGCTACGAGGATGG-3′) targeting the *BoERF1L* coding region was

designed and inserted into a modified pKSE401 CRISPR/Cas9 plasmid to generate *BoERF1L*-CRISPR/Ca9. For *BoERF1L* gene RNAi, a 202-bp specific fragment of the *BoERF1L* coding sequence (44–245 bp) was synthesized and inserted into a pFGC5941 vector to generate a *BoERF1L*-RNAi construct. To generate a *PΔ-597::Ms-cd1* construct, the mutated *Ms-cd1* promoter (amplified from *PΔ-597::gMs-cd1*) and *Ms-cd1* CDS were cloned and inserted into a modified pCAMBIA1301 vector. All plasmids contained *Bar*, *Hyg* or *Kana* resistance genes for selection during *Agrobacterium*-mediated transformation. All plasmids were verified by Sanger sequencing at Tsingke Biological Technology (Beijing, China) and transformed into *Agrobacterium* strain GV3101.

*PΔ-597::gMs-cd1*, *BoERF1L*-RNAi and *BoERF1L*-CRISPR/Ca9 were transformed into the cabbage inbred line 01-20. *Ms-cd1*-CRISPR/Cas9 was used to transform 01-20 and DGMS01-20. *PΔ-597::Ms-cd1* was used to transform *A. thaliana* Col-0, rice ZH11, tomato Ailsa Craig and rapeseed Westar.

*Agrobacterium*-mediated transformation in cabbage, rice, rapeseed, tomato and *A. thaliana* was performed using a standard protocol[39–42]. For genetic transformation in cabbage, hypocotyls from 6-day-old seedlings were cut into 0.8–1 cm explants and precultured for 2 days, and then inoculated and cocultured with *Agrobacterium tumefaciens* strain GV3101 carrying the target vector. After cocultivation for 2 days, the explants were transferred to selection medium (4.43 g/L MS + 0.1 mg/L 1-naphthaleneacetic acid + 1 mg/L 6-bnzylaminopurine + 30 g/L sucrose + 12 mg/L Basta + 300 mg/L Timentin + 0.8% agar; pH: 5.84–5.90). Basta-resistant shoots were obtained under this selection culture. The resulting transgenic plants were identified via PCR amplification in conjunction with primers for *Bar*, *HYG* or *Kana*. At least 5 independent lines were generated for each transformation. To identify the plants with edited genes, the target gene was amplified from genomic DNA and sequenced to search for insertions/deletions within the target site.

## Quantitative real-time PCR (qRT–PCR) analysis

Total RNA was extracted from various cabbage tissues, including roots, stems, leaves, floral buds and anthers, at different stages using an RNAprep Pure Plant Kit (Tiangen, Beijing, China). First-strand cDNA was synthesized using a PrimeScript 1st Strand cDNA Synthesis Kit (Tiangen, Beijing, China) following the protocols described by the supplier. qRT–PCRs were performed using SYBR Premix Ex Taq II (Tli RNase H Plus; Takara) on a CFX96 Touch Real-Time PCR Detection System (Bio-Rad, Hercules, CA, USA). The *B. oleracea Actin* gene (*AFO44573.1*) and *Arabidopsis Actin* gene (*AT5G09810.1*) were used as reference controls. Each sample involved assaying three biological replicates, and there were three technical repeats. The data were analyzed by the $2^{-\Delta\Delta CT}$ method[43], and the relative expression levels are given as the means ± SDs. The primers used for qRT–PCR are listed in Supplementary Data 2.

## Promoter–GUS analysis

The WT promoter (PWT) and mutated promoter (PΔ-597) of *Ms-cd1* were amplified from 01-20 and HO-DGMS01-20, respectively, inserted into a modified pCAMBIA1301 vector in place of the CaMV 35S promoter and fused to a *GUS* reporter gene. The generated *PWT::GUS* and *PΔ-597::GUS* plasmids were verified via Sanger sequencing and transformed into *A. thaliana*. The transgenic plants were germinated on 1/2-strength Murashige and Skoog media supplemented with 30 mg/L hygromycin B, and the presence of the *HYG* gene was verified via PCR. At least 20 independent lines were generated for *PWT::GUS* and for *PΔ-597::GUS*. From the T2 generation, ten *PWT::GUS* lines and ten *PΔ-597::GUS* lines were randomly selected for histochemical *GUS* staining. *Arabidopsis* tissues were fixed in 90% (v/v) acetone and washed with GUS staining buffer (50 mM sodium phosphate buffer, pH 7.2, 0.2% Triton X-100, 2 mM potassium ferrocyanide, and 2 mM potassium

ferricyanide). Then the GUS staining buffer was supplemented with 2 mM X-Gluc (Sigma 203783) until a blue color became visible. The samples were then immersed in 75% (v/v) ethanol to remove chlorophyll. GUS activity was measured by quantitation of 4-methylumbelliferone produced by cleavage of the substrate 4-methylumbelliferyl-b-D-glucuronide (Duchefa) on an F97 pro Fluorometer (Shanghai Lengguang Technology Co., Shanghai, China). qRT–PCR analysis of the *GUS* gene was performed using *Actin* as an internal control.

### RNA in situ hybridization
Floral buds of 01-20 and HE-DGMS01-20 cabbage at different developmental stages were collected, fixed in FAA solution (50% ethanol, 5% acetic acid, 3.7% formaldehyde), and stored at 4 °C overnight. The fixed floral buds were dehydrated through an ethanol solution series of 70%, 80%, 90% and 100% followed by clearing in a xylene solution series of 25%, 50%, 75% and 100% and then embedded in paraffin (Paraplast High Melt, Leica). The samples were cut into 8–10 μm sections using a microtome (RM2255, Leica, Wetzlar, Germany) and then mounted on poly-L-lysine-coated glass slides (P0425-72AE; Sigma–Aldrich). A 200-bp cDNA fragment from *Ms-cd1* was subsequently synthesized and inserted into a pGEM-T Easy vector (Promega, Madison, WI, USA). Digoxigenin (DIG)-labeled sense and antisense probes were produced from T7 and SP6 polymerase using a DIG Northern Starter Kit (12039672910, Roche) following the manufacturer's instructions. RNA ISH with the sense and antisense probes was performed, observed under a BX-51TRF microscope (Olympus, Tokyo, Japan) and imaged with a microcolor charge-coupled device camera (UCMOS05100KPA, ToupTek Photonics, Hangzhou, China).

### Subcellular localization analysis of *Ms-cd1*
To reveal the subcellular localization of *Ms-cd1*, we generated a *35S::Ms-cd1-GFP* transient expression vector. The coding sequence of *Ms-cd1* was fused to the N-terminus of GFP and cloned into and inserted into PC1300. The resulting *35S::Ms-cd1-GFP* vector was then transformed into *Agrobacterium* strain GV3101. *Agrobacterium* suspensions of *35S::Ms-cd1-GFP* and *p19* suppressor were mixed together equally, and infiltration suspensions consisting of 10 mM 2-(N-morpholino)ethanesulfonic acid (MES), 10 mM MgCl$_2$ and 100 mM acetosyringone were prepared. Leaves of tobacco plants were subsequently infiltrated with the suspension. After the leaves were incubated for 18 h in the dark and 24 h under light, the GFP signal in tobacco leaves was excited by a 488-nm laser and detected at 510 nm under a confocal microscope (Carl Zeiss AG, Oberkochen, Germany).

### Phylogenetic analyses
The protein sequences of Ms-cd1 and BoERF1L were used as queries to search for homologs with the BLASTP tool of the NCBI database. Homologous protein sequences were downloaded from the NCBI database, and they were named according to published information or their NCBI accession number. The protein sequences were aligned by ClustalW, and a neighbor-joining phylogenetic tree was constructed with MEGA software (version 7) with the Poisson model and 1000 bootstrap replicates. Information concerning the homologous proteins is shown in Supplementary Data 3.

### Y1H assays
Cabbage floral buds (<3.5 mm) were collected for cDNA library construction via a Matchmaker GAL4 One-Hybrid System (Clontech) following the manufacturer's instructions. A 66-bp fragment (624–559 upstream of ATG) from the WT promoter was synthesized by staff at BGI Tech (Beijing, China) and subcloned and inserted into a pAbAi vector, and the resulting pABAi-WTpro bait was transformed into the yeast strain Y1HGold. Y1H library screening assays were performed against the cDNA expression library using the Matchmaker GAL4 One-Hybrid System (Clontech).

Competent Y1HGold yeast cells containing pABAi-WTpro as bait were transformed with the contents of the cDNA library, plated onto selective SD/-Leu media supplemented with 200 ng/mL AbA, and incubated at 30 °C. The p53-AbAi Control Vector and p53 Control Insert from Clontech were used as the controls. Yeast plasmids from positive colonies were extracted from positive colonies and transformed into *Escherichia coli* DH5α competent cells. The prey fragments from the positive colonies were identified by Sanger sequencing.

To verify the interaction of BoERF1L with the bait, the full-length cDNA sequences of BoERF1L were amplified and fused in frame with the GAL4 activation domain of pGADT7rec (Clontech), forming pGADT7-BoERF1L. In addition to the pABAi-WTpro reporter, we constructed another reporter, pABAi-MTpro, with a 65-bp fragment (624–559 upstream of ATG) from the DGMS01-20 promoter inserted into the pAbAi vector. The pGADT7-BoERF1L construct was subsequently cotransformed with the reporter vectors. DNA–protein interactions were determined according to the growth of the transformants on SD/-Leu media supplemented with 200 ng/mL AbA.

### Electrophoretic mobility shift assay
For expression and purification of the His-BoERF1 recombinant protein, the full-length coding sequence of BoERF1 was fused to a His tag at the C-terminus and then subcloned and inserted into a pET-SUMO vector. The resulting construct was transformed into *E. coli* Rosetta (DE3) cells, and the transformed bacteria were cultured in Luria–Bertani media supplemented with 100 mg/mL *Kana* at 37 °C. Expression of fusion proteins was induced overnight at 30 °C by the addition of 0.1 mM isopropyl b-D-1-thiogalactopyranoside. Recombinant His-BoERF1 was subsequently purified using glutathione-agarose beads (BD Biosciences) according to the manufacturer's instructions.

EMSAs were conducted following the manufacturer's instructions. Briefly, biotin-labeled and unlabeled *Ms-cd1* promoter probes were synthesized by Sangon Biotech (Shanghai, China). The purified BoERF1-His protein and biotin-labeled probes were incubated in a 10 μL reaction mixture that included 2 μg of recombinant protein, 20 fmol of biotin-labeled DNA and 2 μL of 5× EMSA/Gel-Shift Binding Buffer (GS005, Beyotime). The products were analyzed using 6% polyacrylamide gels. To verify binding specificity, 20-, 30- and 40-fold excesses of unlabeled DNA probe were added, and the His-BoERF1 protein was used as a negative control.

### Dual-LUC transient expression assays
The 2.9-kb promoter region of *Ms-cd1$_{PWT}$* was amplified from *PWT::GUS* via PCR and cloned and inserted into a pluc-35Rluc transient expression vector[44] to generate a *PWT::LUC* plasmid used as a reporter. The *BoERF1L* coding sequence was subsequently amplified via PCR from the *35S::Ms-cd1* vector and cloned and inserted into a pCAMBIA 2300 vector to generate the *35S::BoERF1L* effector plasmid. An empty pluc-35Rluc construct was used as a negative control. The plasmids were transformed into *A. tumefaciens* strain EHA105. *Agrobacterium* cells carrying the effector, reporter and p19 plasmids were suspended using 10 mM MgCl$_2$ and 150 μM acetosyringone and mixed in a volume ratio of 2:1:3. The suspensions were infiltrated into leaves of 5–6-week-old tobacco plants using a syringe. The leaves were sampled 48 h after infiltration. Firefly luciferase and Renilla luciferase activities were determined using a dual-LUC reporter assay system (E1910, Promega). The results are shown as *LUC/REN* ratios. Luciferase luminescence was captured using a Tanon 5200 system (Berthold, Tanon, Shanghai, China).

### Reporting summary
Further information on research design is available in the Nature Portfolio Reporting Summary linked to this article.

## Data availability

The genomic resequencing data generated in this study have been deposited in the NCBI Sequence Read Archive under accession PRJNA1013233. The sequence of *BoERF1L* can be found in the Brassica Database (http://brassicadb.cn) by searching ID Bol028757. The sequences of *Ms-cd1_{PWT}* and *Ms-cd1_{PA-597}* have been deposited in Gen-Bank under accessions OR523690 and OR523691, respectively, which are also provided in Supplementary Data 4. Source Data are provided with this paper.

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

## Acknowledgements

This work was supported by grants from the National Natural Science Foundation of China (32072570 to H.L.), the Science and Technology Innovation Program of the Chinese Academy of Agricultural Sciences (CAAS-ASTIP-IVFCAAS to L.Y. and Y.Z.) and the China Agriculture Research System of MOF and MARA (CARS–23 to Z.F. and Y.Z.). We thank Prof. Xia Cui from the Institute of Vegetables and Flowers, Chinese Academy of Agricultural Sciences for kindly providing the vectors of the Dual-Luc assay.

## Author contributions

H.L., Z.F. and F.H. conceived and designed the work. K.Y., F.H. and W.S. performed the experiments. F.H. and K.Y. wrote and revised the manuscript. W.S., X. Zhang, X.L., X. Zhao, L.Y., Y.W., J.J., Y.L., Z.L., J.Z., C.Z., S.H. and Y.Z. analyzed the data and revised the manuscript. All authors have read and approved the final version of the manuscript.

## Competing interests

The authors declare no competing interests.
