## [Peer Review File · Nature Communications]

A natural mutation in the promoter of Ms-cd1 causes dominant male sterility in Brassica oleraceaReviewers' Comments:

Reviewer #1:

Remarks to the Author:

The manuscript entitled "A rare 1-bp deletion in the promoter of Ms-cd1 causes dominant male sterility in 2 Brassica oleracea" by Han et al. described the cloning of a dominant male sterile gene, which encodes a transcription factor. Its orthologs in several other species have been shown to be associated with pollen development. Their loss-of-functions led to male sterile in a recessive manner. Unlike the male sterile in other species, the male sterile investigated by Han et al. is a dominant one, caused by a 1-bp deletion in its promoter region. The authors showed that the 1-bp deletion was disabled the binding of a repressor transcription factor ERF1L. The 1-bp promoter deletion consistently led to a strong promoter in several species including tomato and rice. If I understand it correctly, the results suggest that the binding motif of ERF1L and its repression on Ms-cd1 (PHD-finger) are conserved in different plant species. The manuscript is well prepared, easy to read. The results are sound, provide new insights on dominant male sterile, and can be applied in breeding programs in Brassica oleracea and potentially in other crops as well.

The authors also suggested that the Ms-cd1 gene can be applied to other species. As shown by the manuscript, transformation of the dominant male sterile allele led to no pollen in other species. However, such a strategy is not feasible in breeding programs right now or in the foreseeable future due to the GMO issue. The authors also implied that a base-editing strategy may be used, but it requires a conserved suppression motif in the promoter. The ERF1L gene, on the other hand, may provide an ideal approach to generate a dominant male sterile material without the GMO issue. Knockout of the ERF1L gene should lead to male sterile if the proposed mechanism is correct. The authors knocked down the ERF1L gene, and the mutant showed decreased fertility. I recommend the authors to knock out the ERF1L gene, and to investigate whether such a mutant is dominant male sterile which can be used in hybrid breeding programs.

Minor comments:

L112: Dominant...plants◇The Ms-cd1 dominant...plants DGMS01-20

L113: According to that◇Like in

L135: completely◇morphologically. In all other aspects◇expect pollen development

L159: The expression◇High expression. Delete "native"

L161: 6028◇6,028

L162: 2913◇2,913

L165: Only seven lines and all of them were sterile? Usually, not all lines showed the same phenotype.

L173: is referred to ◇is referred to as

L196: ;◇,

L198: the release◇the release of microspores

L206: What is the phenotype of the F1 hybrid Ms-cd1PΔ-597/ms-cd1PΔ-597?

L228: Did the 1-bp deletion alter its expression pattern?

L241: analyzed◇obtained? Originated—were derived?

L270: What will be the phenotype of ERF1L knockout mutants?

L286: Progeny◇Progenies

L287: according to◇with

L324: and◇or

Reviewer #2:

Remarks to the Author:

Ms-cd1 is a dominant male-sterile line discovered in 1979 which is widely used for cabbage breeding. In this work, Han et al., has successfully identified the Ms-cd1 gene. It is interesting that the candidate gene is PHD-finger motif homologous to MS1 in Arabidopsis, PTC1/OsMS1 in rice, and

ZmMs7 in maize. MS1 is extensively studied during anther development. Its knockdown/out leads to TGMS phenotype in rice and male sterile in all plants investigated. However, a 1-bp deletion in the transcriptional regulatory region of the promoter of Ms-cd1 leads to overexpression and dominant genic male sterility. Furthermore, an ethylene response factor (ERF) BoERF1L was found to directly bind to the promoter of Ms-cd1PWT to repress Ms-cd1 expression. The genetic data is very solid in this work which clearly demonstrates the mechanism of DGMS phenotype.

Although both overexpression and knockout of Ms-cd1 leads to male sterile, the molecular and cellular mechanisms are quite different. I would like to know how overexpression of this gene leads to male sterile. The ectopic expression of Ms-cd1P \otimes -597 conferred DGMS to both dicotyledonous and monocotyledonous plant species, including rice, rabiopsis, tomato, and rapeseed. In Arabidopsis, the genetic pathway DYT1-TDF1-AMS-MS188-MS1 is critical for tapetum development and pollen formation. The expression analysis of this pathway and their downstream genes in P \otimes -597::Ms-cd1 (Arabidopsis) plant may provide clues of the DGMS mechanism.

Reviewer #3:

Remarks to the Author:

Fengqing Han et al. reported the map-based cloning of the dominant male-sterile gene Ms-cd in cabbage, which encodes a transcription factor homologous to MS1 in Arabidopsis, PTC1/OsMS1 in rice, and ZmMs7 in maize. Ethylene response factor (ERF) BoERF1L directly bound to the GAC-core motif in the promoter and repressed the expression of Ms-cd. The 1-bp deletion in the Ms-cd1P-597 prevented this binding ability and enhanced the promoter activity, leading to severe pollen wall defects and conferring dominant genic male sterility (DGMS) to Brassica oleracea. The authors found that a rare 1 bp deletion in the Ms cd1 promoter causes male sterility, but more data is needed to support their conclusions.

The authors observed that P 597 has higher promoter activity, and concluded that the expression level, but not the tissue-specific expression pattern, of Ms cd1 is altered in DGMS plants. I have not found data to support this conclusion.

Both knocking out and enhancing Ms-cd1 leads to male sterility, the mechanism remains unknown. This point can be discussed.

The dual LUC assay showed that the BoERF1L repressed PWT::LUC reporter transcription, but more evidence is expected to show that BoERF1L regulates Ms cd1 expression.

Please show evidence that RNAi 4 and RNAi 6 are RNAi transgenic plants.

BoERF1L as an ERF transcription factor should respond to plant hormones and pathogen attacks and abiotic stresses. So male sterility/fertility will be sensitive to environmental stress.

In the result section lines 278-279: A promising DGMS system for heterosis utilization in dicotyledonous and monocotyledonous crop species. In the example shown in Figure 6E, they didn't show the data of heterosis. I also didn't see the results in monocotyledonous crop species.

Reviewer #1 (Remarks to the Author):

The manuscript entitled “A rare 1-bp deletion in the promoter of *Ms-cd1* causes dominant male sterility in *Brassica oleracea*” by Han et al. described the cloning of a dominant male sterile gene, which encodes a transcription factor. Its orthologs in several other species have been shown to be associated with pollen development. Their loss-of-functions led to male sterile in a recessive manner. Unlike the male sterile in other species, the male sterile investigated by Han et al. is a dominant one, caused by a 1-bp deletion in its promoter region. The authors showed that the 1-bp deletion disabled the binding of a repressor transcription factor ERF1L. The 1-bp promoter deletion consistently led to a strong promoter in several species including tomato and rice. If I understand it correctly, the results suggest that the binding motif of ERF1L and its repression on *Ms-cd1* (PHD-finger) are conserved in different plant species. The manuscript is well prepared, easy to read. The results are sound, provide new insights on dominant male sterile, and can be applied in breeding programs in *Brassica oleracea* and potentially in other crops as well.

The authors also suggested that the *Ms-cd1* gene can be applied to other species. As shown by the manuscript, transformation of the dominant male sterile allele led to no pollen in other species. However, such a strategy is not feasible in breeding programs right now or in the foreseeable future due to the GMO issue. The authors also implied that a base-editing strategy may be used, but it requires a conserved suppression motif in the promoter. The ERF1L gene, on the other hand, may provide an ideal approach to generate a dominant male sterile material without the GMO issue. Knockout of the ERF1L gene should lead to male sterile if the proposed mechanism is correct.

The authors knocked down the ERF1L gene, and the mutant showed decreased fertility. I recommend the authors to knock out the ERF1L gene, and to investigate whether such a mutant is dominant male sterile which can be used in hybrid breeding programs.

Response: Thanks for the comments. We tried our best to knockout of the *BoERF1L* gene during the past two years. We obtained T0 gene editing lines in 2021, and T1 generations in 2022 (cabbage produces only one generation per year, because it needs strict and long-time conditions for vernalization). We completed the genetic and phenotypic analyses of the *boerf1l* mutant in June 2023.

The *erf1l* mutant showed dramatically reduced male fertility, but not completely sterile (these results have been added in Lines 283-289, and shown in Figure 5 and Supplemental Figure 11; and the RNAi results were moved to Supplementary Figure 12). We speculate that perhaps other ERF factor functions redundantly to regulate *Ms-cd1* gene (we have conducted phylogenetic analysis of *BoERF1L*, and found some paralogs, shown in Supplemental Figure 9). The reduced male fertility of *erf1l* mutant is recessive, because that only in the homozygous *boerf1l* mutant, *Ms-cd1* gene is not repressed by *BoERF1L*.

Obviously, it's unlikely to generate a dominant male sterile material by knockout of the *BoERF1L*. We think that, for hybrid breeding propose, dominant male sterile lines can be created by manipulate the promoter region of *Ms-cd1* gene, rather than to knock out the *BoERF1L* gene. It requires precise editing of the specific target in the *Ms-cd1* promoter region. This may be available in the future, with the development of genome editing technology.

Minor comments:

L112: Dominant...plants◇The *Ms-cd1* dominant...plants DGMS01-20

Response: Thanks for the comments. We have revised it.

L113: According to that◊Like in

Response: Thanks for the comments. We have revised it.

L135: completely◊morphologically. In all other aspects◊expect pollen development

Response: Thanks for the comments. We have revised it.

L159: The expression◊High expression. Delete “native”

Response: Thanks for the comments. We have revised it.

L161: 6028◊6,028

Response: Thanks for the comments. We have revised it.

L162: 2913◊2,913

Response: Thanks for the comments. We have revised it.

L165: Only seven lines and all of them were sterile? Usually, not all lines showed the same phenotype.

Response: Thanks for the comments. We obtained 11 independent transgenic lines in total. Among them seven lines were completely male sterile, three lines were semi-sterile, and the remaining one line was indistinguishable from the wild type. We think that this is related to the expression levels of the transgene.

L173: is referred to ◊is referred to as

Response: Thanks for the comments. We have revised it.

L196: ;◊,

Response: Thanks for the comments. We have revised it.

L198: the release◊the release of microspores

Response: Thanks for the comments. We have revised it.

L206: What is the phenotype of the F1 hybrid Ms-cd1PΔ-597/ms-cd1PΔ-597?

Response: The hybrid *Ms-cd1_{PA-597}/ms-cd1_{PA-597}* showed a phenotype identical to DGMS.

L228: Did the 1-bp deletion alter its expression pattern?

Response: Thanks for the comments. In addition to the *in situ* hybridization assays in the last version, we have showed qRT-PCR and GUS expression pattern in various tissues of cabbage lines and Arabidopsis promoter-GUS transgenic lines in Figure 4, which suggested that the 1-bp deletion only causes changes of expression levels.

L241: analyzed◊obtained? Originated—were derived?

Response: Thanks for the comments. We have revised it.

L270: What will be the phenotype of ERF1L knockout mutants?

Response: Thanks for the comments. We have answered this question above. The phenotype of *erf1l* mutant is shown in Figure 5 and Supplemental Figure 11.

L286: Progeny◊Progenies

Response: Thanks for the comments. We have revised it.

L287: according to◊with

Response: Thanks for the comments. We have revised it.

L324: and◊or

Response: Thanks for the comments. We have revised it.

Reviewer #2 (Remarks to the Author):

Ms-cd1 is a dominant male-sterile line discovered in 1979 which is widely used for cabbage breeding. In this

work, Han et al., has successfully identified the Ms-cd1 gene. It is interesting that the candidate gene is PHD-finger motif homologous to MS1 in Arabidopsis, PTC1/OsMS1 in rice, and ZmMs7 in maize. MS1 is extensively studied during anther development. Its knockdown/out leads to TGMS phenotype in rice and male sterile in all plants investigated. However, a 1-bp deletion in the transcriptional regulatory region of the promoter of Ms-cd1 leads to overexpression and dominant genic male sterility. Furthermore, an ethylene response factor (ERF) BoERF1L was found to directly bind to the promoter of Ms-cd1PWT to repress Ms-cd1 expression. The genetic data is very solid in this work which clearly demonstrates the mechanism of DGMS phenotype.

Although both overexpression and knockout of Ms-cd1 leads to male sterile, the molecular and cellular mechanisms are quite different. I would like to know how overexpression of this gene leads to male sterile. The ectopic expression of Ms-cd1P Δ -597 conferred DGMS to both dicotyledonous and monocotyledonous plant species, including rice, Arabidopsis, tomato, and rapeseed. In Arabidopsis, the genetic pathway DYT1-TDF1-AMS-MS188-MS1 is critical for tapetum development and pollen formation. The expression analysis of this pathway and their downstream genes in P Δ -597::Ms-cd1 (Arabidopsis) plant may provide clues of the DGMS mechanism.

Response: Thanks for the comments. We investigated the expression of these genes involved in the *DYT1-TDF1-AMS-MS188-MS1* network. Our result indicated that *Ms-cd1* plays a role in negative feedback regulation of the *BoDYTI-BoTDF1-BoAMS-BoMS188-Ms-cd1* module. In DGMS, promoter mutation causes dysregulation of *Ms-cd1* and strong negative feedback of the *BoDYTI-BoTDF1-BoAMS-BoMS188-Ms-cd1* module, and subsequently down-regulation of pollen wall development related genes such as *BoQRTs*, *BoTEK* and *BoPKSA*. In consistent with the down-regulation of *BoQRTs*, *BoTEK*, DGMS showed some key defects similar to Arabidopsis *qrt* mutant (microspores arrested in tetrads by the undegradable tetrad wall) (Preuss et al. 1994; Francis et al. 2006), and *tek* mutant (absence of nexine and intine layers of microspores) (Lou et al. 2014). In addition, we proposed a working model of Ms-cd1 regulating male fertility. These information have been added in Results section Lines 302-333 and shown in Supplementary Figure 13.

References:

- Preuss D, Rhee S Y, Davis R W. Tetrad analysis possible in Arabidopsis with mutation of the *QUARTET (QRT)* genes. *Science*, 1994, 264(5164): 1458-1460.
- Francis K E, Lam S Y, Copenhaver G P. Separation of Arabidopsis pollen tetrads is regulated by *QUARTET1*, a pectin methylesterase gene. *Plant physiology*, 2006, 142(3): 1004-1013.
- Lou Y, Xu X F, Zhu J, et al. The tapetal AHL family protein TEK determines nexine formation in the pollen wall. *Nature Communications*, 2014, 5(1): 3855.

Reviewer #3 (Remarks to the Author):

Fengqing Han et al. reported the map-based cloning of the dominant male-sterile gene Ms-cd in cabbage, which encodes a transcription factor homologous to MS1 in Arabidopsis, PTC1/OsMS1 in rice, and ZmMs7 in maize. Ethylene response factor (ERF) BoERF1L directly bound to the GAC-core motif in the promoter and repressed the expression of Ms-cd. The 1-bp deletion in the Ms-cd1P-597 prevented this binding ability and enhanced the promoter activity, leading to severe pollen wall defects and conferring dominant genic male sterility (DGMS) to Brassica oleracea. The authors found that a rare 1 bp deletion in the Ms cd1 promoter causes male sterility, but more data is needed to support their conclusions.

The authors observed that P 597 has higher promoter activity, and concluded that the expression level, but not the tissue-specific expression pattern, of *Ms cd1* is altered in DGMS plants. I have not found data to support this conclusion.

Response: Thanks for the comments. In addition to the in situ hybridization assays and GUS results in the last version, we have added qRT-PCR and GUS expression pattern in various tissues of cabbage and *Arabidopsis* promoter-GUS transgenic lines (in Lines 221-242, and shown in Figure 4).

We have to declare that, after multiple independent assays in different cabbage backgrounds, we found that the expression levels of *Ms-cd1* (*ms-cd1*) is quite weird when detected by qRT-PCR. In contrast to the result of GUS assays, qRT-PCR assays showed that *Ms-cd1* is significantly down regulated in DGMS01-20 compared with 01-20. And intriguingly, in consistent with the GUS assays results, qRT-PCR assays showed that the expression levels of *ms-cd1* in *ms-cd1*_{PA-597} mutant is about 3 fold-higher than that in *ms-cd1*_{PWT}. Additionally, the *ms-cd1* (or *Ms-cd1*) expression in *ms-cd1* mutants and is significant higher than that in *Ms-cd1*_{PWT} and *Ms-cd1*_{PA-597} plants, suggesting that the functional *Ms-cd1* repressed the expression of itself.

Self-repression of *MSI* (the orthologous of *Ms-cd1* in *Arabidopsis*) is also reported in *Arabidopsis*. Yang et al. reported that ‘an increase of *ms1* transcript is detected in the *ms1* mutant, implying that self-regulatory feedback, based upon a functional *MSI* transcript, is limiting wildtype *MSI* expression’. In addition, in a recent study, Hou reported that *ZmMs7* (the orthologous of *Ms-cd1*) in maize involved in a feedback repression loop (*ZmHHLH51-ZmMYB84-ZmMS7-ZmMS1*), in which *ZmMS7* activates *ZmMs1/ZmLBD30* transcription, and in turn *ZmMS1/ZmLBD30* represses all the three upstream activators. Therefore, we speculated that transcriptional ability of *Ms-cd1* cannot be accurately reflected by qRT-PCR result when a functional *Ms-cd1* is present.

References:

Yang C, Vizcay-Barrena G, Conner K, et al. *MALE STERILITY1* is required for tapetal development and pollen wall biosynthesis. *The Plant Cell*, 2007, 19(11): 3530-3548.

Hou Q, An X, Ma B, et al. *ZmMS1/ZmLBD30*-orchestrated transcriptional regulatory networks precisely control pollen exine development. *Molecular Plant*, 2023.

Both knocking out and enhancing *Ms-cd1* leads to male sterility, the mechanism remains unknown. This point can be discussed.

Response: Thanks for the comments. We investigated the expression of genes involved in the *DYT1-TDF1-AMS-MS188-MS1* network, and found that both *Ms-cd1*_{PA-59} and loss of function of *Ms-cd1* alters expression of a series of genes required for tapetum and pollen development. Most of them showed contrary expression levels in DGMS01-20 and *ms-cd1*. Our results indicated that *Ms-cd1* involved in a negative feedback regulation of the *BoDYT1-BoTDF1-BoAMS-BoMS188-MS-cd1* module, and the precise regulation of *Ms-cd1* is vital for male fertility. In DGMS, promoter mutation causes strong negative feedback of the *BoDYT1-BoTDF1-BoAMS-BoMS188-MS-cd1* module, and subsequently down-regulation of pollen wall development related genes such as *BoQRTs*, *BoTEK* and *BoPKSA*. In RGMS, *ms-cd1* cannot feedback the module, which result in upregulation of some genes such as *BoTEK* and *BoPKSA*, and as well as down-regulation of others pollen development genes such as *BoLTPs* and *BoEXLs*. We have added these information in Results and Discussion sections (Lines 302-333, 421-434, and shown in Supplementary Figure 13).

The dual LUC assay showed that the BoERF1L repressed PWT::LUC reporter transcription, but more evidence is expected to show that BoERF1L regulates *Ms-cd1* expression.

Response: Thanks for the comments. We generated *boerf1l* mutant and provide genetic evidence that BoERF1L regulates *Ms-cd1* expression. These results have been added in Lines 283-289, 297-300 and shown in Figure 5 and Supplemental Figure 11; and the RNAi results were moved to Supplementary Figure 12.

We found that in *erf1l* mutant, expression levels of *Ms-cd1* is altered similar to that in the DGMS lines, i.e. down-regulated when detected by qRT-PCR (As described above, due to the negative feedback of the *BoDYT1-BoTDF1-BoAMS-BoMS188-Ms-cd1*, when functional *Ms-cd1* is present, expression levels detected by qRT-PCR is quite weird.). In addition, we detected the expression of genes closely related to *Ms-cd1*, and found that all the detected genes showed altered expression levels identical to that in the DGMS lines. Therefore, as *Ms-cd1* and its related genes showed a highly similar expression changes between *erf1l* mutant and DGMS plants, its highly probably that BoERF1L regulates *Ms-cd1* expression, although we cannot say that 'BoERF1L represses *Ms-cd1*'.

Please show evidence that RNAi 4 and RNAi 6 are RNAi transgenic plants.

Response: Thanks for the comments. We have provided PCR detection results using specific primers for *Bar* gene, and the expression levels of *BoERF1L* (Supplementary Figure 12), which showed that RNAi 4 and RNAi 6 are *BoERF1L* RNAi transgenic plants.

BoERF1L as an ERF transcription factor should respond to plant hormones and pathogen attacks and abiotic stresses. So male sterility/fertility will be sensitive to environmental stress.

Response: Thanks for the comments. The performance of *Ms-cd1* male sterility/fertility seems quite complex depending on genetic background. In some background, this DGMS is unstable and believed to be sensitive to low temperature (Lou et al. 2007; Ji et al. 2020). However, in some background, the male sterility is quite stable.

References:

Ji J, Huang J, Yang L, et al. Advances in Research and Application of Male sterility in Brassica oleracea. Horticulturae, 2020, 6(4): 101.

Lou P, Kang J, Zhang G, et al. Transcript profiling of a dominant male sterile mutant (*Ms-cd1*) in cabbage during flower bud development. Plant Science, 2007, 172(1): 111-119.

In the result section lines 278-279: A promising DGMS system for heterosis utilization in dicotyledonous and monocotyledonous crop species. In the example shown in Figure 6E, they didn't show the data of heterosis. I also didn't see the results in monocotyledonous crop species.

Response: Thanks for the comments. In Figure 6, we showed a proposed system for heterosis utilization in dicotyledonous and monocotyledonous crops. We added the data of heterosis for cabbage hybrid Zhonggan21 produced via this DGMS-based system (Supplementary Figure 14). However, this system has not been tested in monocotyledonous crop species. Therefore, we have removed the Figure 6F and G.

Reviewers' Comments:

Reviewer #1:

Remarks to the Author:

The authors addressed all my concerns well. I am glad to know that they have started to knock out the suppressor several years ago and got the results in time.

Reviewer #2:

Remarks to the Author:

In this revision, Han et al., performed further analysis to understand a 1-bp deletion in the promoter of Ms-cd1P~~597~~ leading to pollen wall defects and male sterility in Brassica oleracea. They demonstrated the strong negative feedback of the BoDYT1-BoTDF-BoAMS-BoMS188-Ms-cd1 module which further affected several genes for pollen formation. In the DGMS line, in addition to enhanced promoter activity, its gene expression may also be advanced. This is for the author's reference.

I have no further questions.

Reviewer #3:

Remarks to the Author:

The manuscript by Han et al. reported that a rare 1 bp deletion in the Ms cd1 promoter causes male sterility. The authors have addressed my concerns and additional data have been included in the revised manuscript. I propose to accept the manuscript.

Reviewer #1 (Remarks to the Author):

The authors addressed all my concerns well. I am glad to know that they have started to knock out the suppressor several years ago and got the results in time.

Response: Thank you. We are happy to hear about this.

Reviewer #2 (Remarks to the Author):

In this revision, Han et al., performed further analysis to understand a 1-bp deletion in the promoter of *Ms-cd1_{P Δ -597}* leading to pollen wall defects and male sterility in *Brassica oleracea*. They demonstrated the strong negative feedback of the BoDYT1-BoTDF-BoAMS-BoMS188-*Ms-cd1* module which further affected several genes for pollen formation. In the DGMS line, in addition to enhanced promoter activity, its gene expression may also be advanced. This is for the author's reference.

I have no further questions.

Response: Thank you. We agree with this comment. The *Ms-cd1_{P Δ -597}* may produce more *Ms-cd1* transcripts in a possible transient point.

Reviewer #3 (Remarks to the Author):

The manuscript by Han et al. reported that a rare 1 bp deletion in the *Ms cd1* promoter causes male sterility. The authors have addressed my concerns and additional data have been included in the revised manuscript. I propose to accept the manuscript.

Response: Thank you. We are happy to hear about this.